# Effects of competition and ownership on the simultaneous relationship between bank risk and capital: Evidence from an emerging economy of Bangladesh

Changjun Zheng[1], Md Mohiuddin Chowdhury[2]*, Anupam Das Gupta[2‡], Md Nazmul Islam[3‡]

1 School of Management, Huazhong University of Science and Technology, Wuhan, P.R. China,
2 Department of Finance, University of Chittagong, Chattogram, Bangladesh, 3 Department of Accounting, Mawlana Bhashani Science and Technology University, Santosh, Tangail, Bangladesh

☯ These authors contributed equally to this work.
‡ ADG and MNI also contributed equally to this work.
* mohiuddincu@cu.ac.bd

**Data Availability Statement:** All relevant data are within the paper and its Supporting Information files.

## Abstract

This study aims to investigate the simultaneous relationship between bank risk and capital, specifically examining how competition and ownership jointly influence this relationship. We employed the two-step system generalized method of moments to address concerns regarding endogeneity and unobserved heteroscedasticity. Based on data from 44 commercial banks in Bangladesh from 2010 to 2021, our findings reveal several key insights: (I) There is a significant non-linear bidirectional relationship between bank risk and capital: capital exhibits a U-shaped effect on risk, while risk has a non-linear negative impact on capital; (II) in highly contested markets, banks hold higher levels of capital, and heightened competition reduces the risk appetite of commercial banks; while less competition has the opposite effect; (III) private and Islamic commercial banks are more risk-averse than state-owned and conventional ones, and (IV) Private and Islamic banks with strong capital positions in a competitive market can better manage risks than state-owned and conventional banks. However, maintaining higher capital ratios in competitive environments is more challenging for state-owned and Islamic banks than private-owned and conventional commercial banks. These results align with the moral hazard hypothesis, competition fragility hypothesis, and the political view of state ownership. Significant insights from the study will fuel the regulators in shaping policies and regulations surrounding emerging countries like Bangladesh, particularly regarding risk, capital, competition, and ownership policies.

## 1 Introduction

Bank failures can cause a domino effect that harms the financial system, making it a significant concern for bank safety and stability. Financial distress befalls when banks lose their ability to

**Funding:** The author(s) received no specific funding for this work.

satisfy financial obligations. The worldwide financial crisis (GFC) and aftermath of the GFC of 2007-08 are evidence of the severe consequences for banks and their customers [1]. As a result, regulators across the globe have made considerable efforts to promote a stable banking system [2]. The revision of the Basel Accord and its worldwide adaptation evidenced the concern of regulators in this regard. For instance, Basel II was adopted in January 2008 to ensure supervisory scrutiny and market discipline, and Basel III, from 2010 onwards improved the quality and quantity of mandatory bank capital [3]. Escalation of capital is a costly and time-consuming process. However, regulatory compliance, empirical evidence of effective risk-mitigating tools, and enhancing stability fuel banks to escalate their capital exposure. As a result, banks, as financial intermediaries, prioritize risk aversion, reduce credit lines, and increase cash reserves [4].

Although regulatory capital requirements can benefit economic well-being by encouraging banks to invest in safer projects, they can also alter the competitive landscape of the banks. Higher regulations can limit entry into the sector and amplify the price-setting power of existing banks, which can have adverse effects if not considered with the characteristics and competition of the banking system, particularly in undercapitalized sectors [5]. Furthermore, lowering capital requirements can hinder smaller banks, as large banks can increase deposit rates and specialize in different types of loans, resulting in intense competition that can drive smaller banks to take on more risk [6], ultimately reducing financial stability. The relationship between competition and banks' risk-taking is complex, with mixed evidence [7]. However, Increased competition can also decrease a bank's charter value, exacerbating the agency problem [8].

Escalation of capital leads banks to charge more interest on loans and other assets to protect the cost of capital. Consequently, increasing the cost of funding and lending enhances the risk that affects banks' profit [9–11]. Agency theory postulates that information diversity and more access to information lead managers to prioritize their interests over wealth maximization [12]. It is argued that ownership structure is vital in allocating value among equity owners [13]. It can also help mitigate agency problems by enabling owners to monitor agent activities actively [14]. However, as risk-averse economic agents, banks prioritize profit maximization and enhancing shareholders' value according to the dealership theory of financial intermediation [15]. The value distribution among equity owners is a defining feature of ownership structure. A well-structured ownership can help address agency issues as owners closely oversee agent activities [14]. Ownership structure and agency costs influenced the capital structure of the banks [16]. Therefore, it is increasingly important to study how ownership and competition impact the bidirectional relationship between risk and capital to gain valuable insights.

Recent studies highlight a research gap in examining the effects of ownership and competition on the simultaneous relationship between risk and capital based on the commercial banks of Bangladesh. While Moudud-Ul-Huq et al. [17] investigated the impact of ownership structure on the relationship between risk and capital, they did not consider the effects of competition. Similarly, Zheng et al. [18] also overlooked the effects of competition. Another study by Mia [19] explored the effect of competition and capital regulation on the financial intermediation cost of commercial banks in Bangladesh. In this quest, this paper aims to address four main questions to investigate the joint effect of ownership and competition on the relationship between bank risk and capital by applying the two-step system generalized methods of moments (TSGMM) related to an emerging economy like Bangladesh, that have not been considered in the previous literature (e.g., [19–22]): (i) How do capital and risk mutually impact each other in the simultaneous relationship? (ii) Does ownership play a significant intermediation role in this relationship? (iii) Does competition play a significant role in this relationship? (iv) Do ownership and competition jointly intermediate the relationship between risk and capital? In light of the issues outlined in the research background, the study's primary aim is to

investigate the impact of competition and ownership on the simultaneous association between bank risk and capital for commercial banks in Bangladesh.

This article significantly contributes to the existing related literature in the following ways. Firstly, to our knowledge, it introduces a novel approach by employing the TSGMM to explore how competition and ownership jointly affect the simultaneous relationship between bank risk and capital. This study differs from prior studies, such as [17–19], where earlier studies mainly focused on the impact of ownership or competition, examining the relationship between risk and capital. However, unlike previous studies, this study adds a novel perspective to the existing body of research. Secondly, the study explores the bidirectional non-linear relationship between bank risk and capital, shedding light on the intricate dynamics of banking systems. Thirdly, we incorporate the interim variable by trading off competition, capital, ownership, and risk, which depicts a clear insight into the relationship status of risk and capital in a competitive market environment with ownership diversity. Fourthly, methodologically, robustness checks using two-stage least squares (2SLS) methods ensure the reliability of findings alongside various specifications of dependent and main independent variables. Fifthly, this study evaluates the reinforcement of various theories, including agency theory, regulatory hypothesis, moral hazard hypothesis, competition fragility hypothesis, charter value view of competition, and political view of state ownership. This study delves empirically into applying these theories in trading off between risk, capital, competition, and ownership in the context of an emerging Asian economy. Lastly, the study's findings not only hold relevance for the specific emerging Asian economy studied but also offer broader implications for similar economies facing analogous economic circumstances, thus contributing significantly to academic discourse.

The remaining part of the paper is structured as follows: Section 2 discusses the related literature and the development of hypotheses. Section 3 outlines the data, variables, and methodology. Section 4 describes the empirical findings and analysis. Lastly, Section 5 concludes the paper.

## 2 Literature review

This section is dedicated to an in-depth exploration of the relevant literature. We begin with the theoretical and conceptual foundations to establish a comprehensive understanding. We then delve into the review of empirical investigations, providing practical insights. We then delve into the theoretical and conceptual foundations to establish a comprehensive understanding.

### 2.1 Theoretical and conceptual background

Most studies examining the risk and capital mentioned the relevance of moral hazard hypotheses between their tradeoff [23]. According to the moral hazard hypothesis, undercapitalized banks respond to moral hazard inducements by increasing loan portfolio risk, ultimately leading to higher non-performing loans [24]. This is because the risk of losing market confidence and reputation forces them to pursue riskier investments to achieve higher returns (e.g., [25]). Capital is one of the prime sources of funds for investment and safeguards against economic shocks [26, 27]. However, incorporating higher capital requirements is accompanied by a slight decrease in lending, whereas lower capital in the banking sector results in an elevated level of systematic risk [28]. According to the regulatory view, for regulatory pressure, e.g., Basel Accord and deposit insurance, banks must shape their capital behavior by holding and managing regulatory capital [29]. The buffer capital view posits that banks must keep excess cash to avoid associated costs [30], and bank-specific characteristics determine the amount of

excess capital over regulatory capital [31]. Thus, the prescription of capital regulation is inconclusive and bank-specific. Again, since the investment mechanism is not the same in all types of banks due to ownership diversification, competition, and ownership become relevant in the risk-capital nexus [32].

According to the charter value view of competition (competition-fragility hypothesis), increasing the tendency of competition causes decreases in the charter value of the bank, which influences banks to take more risks [8, 33]. Moreover, the competition-stability hypothesis proposes that financial stability is enhanced by competition [34]. Again, banks tend to be less competitive in markets with high concentration, as stated by the structure conduct performance hypothesis. This is because having a large market share gives the bank advantages, such as lower collaboration costs, leading to collusive behavior to generate extraordinary profits [35, 36].

Two contrasting perspectives, namely the grabbing hand theory and the helping hand theory, can explain the presence of government ownership. According to the grabbing hand theory, when the government owns a company, it is highly vulnerable to political influences [37, 38], leading to a decline in market efficiency and shareholder value [39]. In contrast, the helping hand theory argues that government intervention can aid a company in remaining viable and enhancing efficiency [40]. Another theory related to state ownership is the network theory, which suggests that firms and governments use economic resources with social and political considerations rather than economic potential and profit, where the possibility of default arises and the firm's risk increases. However, due to the network, firms not only gather financial resources efficiently and promptly but can also stimulate innovation through information sharing [41].

## 2.2 Empirical review and hypothesis development

**2.2.1 Literature regarding risk-taking and capital.** The association between risk-taking and bank capital ratios is theoretically confusing and unclear. Various studies present different evidence concerning the connection between risk and capital. However, capital and risk can positively or negatively influence each other [42].

Abdul Wahab et al. [43] examined the relationship between risk and capital, utilizing variables such as capital adequacy ratio to measure bank capital and used risk-weighted assets to total assets as a proxy of risk-taking. They applied a panel vector error correction model and a two-step dynamic system GMM to analyze panel data from Malaysian commercial banks. Their findings indicate a unidirectional positive relationship between regulatory capital and total risk, suggesting that these banks were likely to engage in riskier projects after meeting regulatory capital requirements. The positive effects of risk on bank capital were also evidenced in the studies of Mateev et al. [44], Alves et al. [45], Mujtaba et al. [46], and de Moraes et al. [47]. Conversely, Moudud-Ul-Huq et al. [17] examined the relationship between capital regulations and risk-taking behavior, using the capital adequacy ratio as a proxy for capital and the ratio of non-performing loans to total loans as a proxy for risk. They employed a two-step system GMM to analyze panel data from Bangladeshi commercial banks and found a negative relationship between capital and risk. Similar negative relationships were also observed in the studies of Anh [48] and Zheng et al. [18]. Additionally, Garel et al. [49] reported mixed results.

The reverse causality of capital in risk is also evident. Arguments also come forward regarding the relevance of capital in managing risk and vice versa. Abbas and Ali [42] advocated capital as a significant determinant in the risk-taking of commercial banks in the USA. The positive effects of capital on risk-taking are also evidenced by the studies of Abdul Wahab et al. [43], Mateev et al. [44], and Drakos et al. [50]. Conversely, Vuong et al. [51], Hogan and

Meredith [52], Gong et al. [53], Cebenoyan and Strahan [54], Anh [48] Zheng et al. [18], Anginer et al. [55], Angkinand and Wihlborg [56] and Laeven and Levine [57], found a negative result of capital on the risk-taking.

However, the association between risk and capital is not straightforward and constant. Empirical evidence also advocates the nonlinear alliance between capital and banks' risk-taking. A study by Jiang et al. [25] showed evidence of the U-shape connection between capital buffers and risk-taking. That refers to the association between risk and capital initially following a negative trend, subsequently depicting a positive association between the two. Chun et al. [58] conducted a study that showed that capital is not a crucial determinant of risk in Korean banks. Conversely, Japanese banks have a significant converse association between capital and risk. The mixed relationship is also evidenced in the studies of Lee and Hsieh [59], Jumreornvong et al. [60], Mateev et al. [61], and Holod et al. [62].

Based on the preceding discussion, we construct the following hypotheses to illustrate the association between risk and capital:

$H_1$: *Capital has a significant negative and non-linear association with Risk and vice versa.*

**2.2.2 Literature related to the impacts of competition on risk-taking and capital.** The issue of competition is complex in the banking sector because increased competition can enhance efficiency but may harm financial stability; however, sometimes, intense competition can be beneficial in achieving the desired level of stability for society [33]. Several scholars have tested the association between competition and risk and capital, utilizing data from banks in single-country or cross-country. These investigations have yielded an array of results and evidence. For example, Hassan Al-Tamimi and Jellali [63] disclosed mixed results regarding the effects of competition on bank risks. The authors observed a diversified relationship between competition and risk in different ownership of banks. Mateev et al. [44] also evidenced a negative association between competition and risk, but their research revealed that the correlation between competition and bank risk is insignificant in Islamic banks. Again, Moudud-Ul-Huq et al. [34] found that less competition reduces banks' credit risk and increases financial stability. The study has also evidenced that Islamic banks in the MENA region display a more robust position regarding ownership structure than specialized government establishments and commercial banks. However, specialized government establishments are riskier than both Islamic and commercial banks. Behr et al. [64] found that capital regulation in the low-concentration market lessened the banks' willingness to take risks. In contrast, the highly concentrated market's relationship between capital and risk is insignificant.

Conversely, positive associations are also observed in the literature. Saif-Alyousfi and Saha [35] conducted a study examining bank competition's impact on risk. They utilized unbalanced panel data of listed commercial banks in GCC countries from 1998 to 2016 and employed two-step system GMM methods. They found that market power decreases bank risk and increases bank stability, while competition and high concentration increase bank risk and decrease bank stability. Greater competition increases risk, which is also evidenced in the study of Srivastava et al. [65], and González [66]. On the other hand, a non-linear association between market concentration and bank risk was evidenced in the study of Mateev et al. [61]. Market competition and concentrations are inversely related [44].

Again, various studies have shown a different relationship between competition and bank capital. While some studies found an inverse effect of competition on bank capital, others found a positive effect. For instance, Zheng et al. [67] investigated the effects of competition on commercial banks' risk, capital, and efficiency in Bangladesh, China, and India. They

utilized a two-step GMM approach, measuring capital with the ratio of eligible capital to total assets, risk with the ratio of non-performing loans to total loans, and competition using the Boone indicator. Their finding demonstrated that competition does not necessarily drive banks to upsurge their capital holdings. Similarly, Soedarmono et al. [20] found a negative relationship between competition and capital, with high market power positively related to bank capital. The positive relationship between market power and capital is also substantiated by Berger et al. [68] study. In contrast, Allen et al. [21] argued that in the absence of deposit insurance, a competitive market structure encourages banks to maintain a positive level of capital, which was supported by Schaeck and Cihák [69] study that found a positive relationship between competition and capital holdings. This positive effect of competition on bank capital was also observed in the research of Berger et al. [70]. Furthermore, Li [71] discovered that banks with higher Tier 1 Capital also exhibit greater market power.

$H_2$: The joint effects of competition and capital (risk) significantly influence risk (capital).

**2.2.3 Literature regarding the effect of ownership on risk-taking and capital.** The ownership structure is an essential determinant of bank performance and corporate governance [72, 73] and helps mitigate disputes among principals and agents [12]. Research evidenced that the ownership structure is a key determining factor of bank risk [74]. Risk-taking behavior differs from bank to bank. There are differing views presented in several research papers regarding the connection between bank ownership and risk. Some studies indicate that ownership concentration leads to higher risk [57, 75, 76], while others suggest that concentrated ownership results in decreased risk [63, 77–79]. Banking system ownership structures in the banking system are more associated with systematic risk than the individual bank's ownership concentration [80]. Several studies have found that private banks are less risky than those owned by the state [81, 82].

Government-owned banks play a crucial role in the financial system of developing countries. Nevertheless, their presence can negatively affect the performance of privately-owned banks [83]. Iannotta et al. [84] exposed that government banks are less capitalized than privately owned ones. Again, in GCC countries, state-owned banks outperform private-owned banks, whereas Islamic banks outperform conventional banks [72]. On the other hand, foreign banks hold higher risk-based capital ratios than domestic ones [85]. Drakos et al. [50] opined that foreign banks generally take higher risks than domestic ones. Lassoued et al. [86] demonstrated a clear and positive correlation between state ownership and capital. Conversely, foreign ownership and capital share a negative association. This indicates that state-owned banks maintain a higher capital level due to their cautious approach. In contrast, foreign banks are limited in risk exposure, resulting in lower capital. Moreover, Zheng et al. [18] investigated the bidirectional relationship between capital regulation and risk-taking in Bangladeshi commercial banks, considering ownership structure, using the two-stage least squares (2SLS) method. Their key findings indicate that active shareholding enhances capital ratios, private banks are more risk-averse than state-owned banks, and Islamic banks outperform conventional banks despite having lower capital stability.

The available literature depicts the apparent presence of ownership in the risk and capital of banks. To investigate the impact of ownership on risk and capital nexus, the relevant hypotheses are:

$H_4$: The joint effects of ownership and capital (risk) significantly influence the risk (capital).

$H_5$: Ownership and competition have a significant joint influence on risk (capital).

# 3 Data, variables, and methodology of the study

## 3.1 Data

We collected bank-level data from audited financial statements of respective banks' websites from 2010 to 2021. Industry-level and macroeconomic data were also obtained from the World Bank dataset. There are 61 scheduled banks in Bangladesh, including 3 Specialized Banks, 6 State-owned Commercial Banks, 9 Foreign Commercial Banks, and 43 Private Commercial Banks ((including 33 conventional PCBs and 10 Islamic Shariah-based PCBs) [32, 87]. Initially, all commercial banks were considered for inclusion in the sample. However, we applied specific criteria to ensure data quality and relevance to our research objectives. We employed a purposive sampling approach, selecting banks based on the following criteria: We excluded Foreign Commercial and Specialized Banks due to inconsistent reporting and non-availability of data. We categorized the remaining banks by ownership (State-owned, Private-owned, Conventional, and Islamic). Banks not producing annual reports for at least five years are excluded to guarantee data quality and remove extreme outlier values from the dataset. After implementing the specified filters, the resultant dataset comprises an unbalanced panel of 494 annual observations spanning 44 commercial banks. Among these, there were 6 SCBs and 38 PCBs. Furthermore, based on the classification of conventional and Islamic banks, the sample included 37 conventional banks, and 7 Islamic banks. The study period spans 12 years and is carefully selected to encapsulate significant economic cycles, regulatory shifts, and data availability crucial to understanding Bangladesh's bank competition and ownership dynamics. This timeframe was chosen for its relevance to the post-global financial crisis era, during which Bangladesh's financial sector underwent notable transformations, marked by the rapid adoption of digitization initiatives, extending banking services even to rural areas [88], and attaining lower-middle income status in 2015. We opted for unbalanced panel data in the empirical study to have maximum degrees of freedom in our quantitative research design.

## 3.2 Description of variables

**3.2.1 Dependent variables.** *Risk measures*. Following the study of Zheng et al. [89] and Srivastava et al. [65], we utilize the widely accepted ratio of non-performing loans to total loans (NPL), a key component of financial statements, to assess banks' risk. To accurately gauge the non-linear impact of risk on bank capital, we incorporated a quadratic term of NPL into our analysis. To further validate our conclusions, we also evaluated the model's resilience in terms of risk by utilizing the ratio of loan loss provision to total loans (LLP) as an additional measure.

*Capital measures*. Following the previous studies [65, 90], we measure capital with capital adequacy ratio; for the robustness test, we use the Tier-1 capital ratio as an alternative measure. We also included a quadratic term of capital for measuring the non-linear effect of capital on bank risk.

**3.2.2 Independent variables.** *Competition*. This study has opted for the Boone indicator as a competition measure following the literature of Mia [19], Gupta and Yesmin [32], and Faia et al. [91]. The Boone indicator measures the degree of competition, calculated as the elasticity of profits to marginal costs. Since the information on Boone is no longer available in the World Bank database after 2017, we empirically determine Boone's value from the aggregate industry data as Schaeck and Cihák [92] did. The model for estimating Boone is:

$$\pi_{it} = \alpha + \beta \ln (MC_{it}) \tag{1}$$

Where $\pi_{it}$ is the profit measure of bank 'i' at the time 't' as measured by return on assets

**Table 1. Description of the variables.**

| Classification | Variable | Description | Literature References/ Sources |
|---|---|---|---|
| Capital | CAR | Capital Adequacy Ratio: the sum of Tier 1 and Tier 2 capital to the total risk-weighted assets. | Zheng et al. [95] |
| | T1C | The ratio of Tier 1 capital to total assets ratio | Abbas and Ali [42] |
| Risk | NPL | The ratio of nonperforming loans to total loan | Zheng et al. [89] |
| | LLP | The ratio of Loan Loss Provision to Total Loan | Davis et al. [96] |
| Competition | Boone | Boone Indicator: $\pi_{it} = \alpha + \beta \ln(MC_{it})$ Where $\beta$ denotes the Boone Indicator, $\pi$ denotes profit measure, and MC is the marginal cost. | Zheng et al. [67]. Authors Calculation |
| Ownership: Dummy variables | POD | Ownership dummy. Equals one (1) for the private-owned banks and zero (0) for the state-owned commercial banks. | Rahman et al. [11]. Authors Calculation |
| | IOD | Ownership dummy. Equals one (1) for the Islamic banks and zero (0) for the conventional banks. | Alqahtani et al. [94]. Authors Calculation |
| Banking Sector Development | BSD | The ratio of industry assets to GDP | Gupta and Yesmin [32] Source: World Bank data |
| Gross domestic product. | GDP | Growth of gross domestic product | Talbi and Bougatef [97] Source: World Bank data |
| Inflation | Inflation | Annual rate of inflation | Claeys and Vander Vennet [98] Source: World Bank data |
| Cost of Financial Intermediation | CFI | The ratio of the difference between interest income and interest expenses to total interest-earning assets. | Gupta et al. [99] |
| Leverage | Lever | Total debt to total assets ratio | Rahman et al. [11] |
| Direct Taxation | DT | Tax to pre-tax profit ratio | Saeed et al. [100] |
| Bank Size | Size | The logarithm of the total deposit | Dwumfour [101] |
| Profitability | ROA | The ratio of net income to total assets | Carsamer et al. [102] |

Source: Author's compilation using the mentioned sources/references.

(ROA), $\beta$ denotes the Boone Indicator. The Boone indicator is inversely related to competition, with more negative values indicating a more competitive market. To establish direct proportionality to competition, we use the opposite of the BI (i.e., $-\beta_t$) by following the study of Tabak et al. [93].

*Ownership*. Following the study of Rahman et al. [11] and Alqahtani et al. [94], we categorize banking institutions into four distinct groups: state-owned commercial banks (SOCBs), Private commercial banks (PCBs), conventional commercial banks (CCBs), and Islamic commercial banks (ICBs). Table 1 presents a comprehensive description of variables used in the study using the mentioned sources and references noted in the last column of the table.

### 3.3 Empirical research framework

This part describes the research methodology and empirical framework covering the baseline and extended bank risk and capital model. We conduct the DWH (Durbin-Wu-Hausman) test to check for endogeneity issues in the model (see, e.g., [103–106]). The result from DWH test statistics demonstrates that the bank risk measures (NPL and LLP) and capital measures (CAR and T1C) are endogenous and present considerable challenges in our model. Therefore, using the OLS (Ordinary Least Square) method to obtain results cannot be consistent. Furthermore, due to heteroscedasticity (White test for heteroscedasticity) and autocorrelation (Breusch-Godfrey LM test), the results of the pre-diagnostic tests also advise not to employ OLS as a regression method (See Appendix C in S1 Appendix). Again, we opted for the simultaneous equations approach to handle the endogeneity issue between capital and risk because

this framework style effectively controls the endogeneity problem [107]. Moreover, the findings of Granger Causality Tests (GCT) for CAR and NPL suggest a significant two-way causal relationship between bank risk and capital, as presented in Table A1 in Appendix A in S1 Appendix.

So, following the studies of Bagh et al. [108], Bagh et al. [109], Moudud-Ul-Huq et al. [34], Akhtar et al. [110], Akhtar et al. [111], Saif-Alyousfi and Saha [35], Gupta et al. [99], Boulanouar et al. [112], and Soedarmono and Tarazi [113], this study utilized the two-step system generalized method of moments (TSGMM) of Arellano and Bond [114] and Arellano and Bover [115]. The TSGMM model controls for endogeneity by internally transforming the data and including the dependent variables' lagged values. The TSGMM application has been identified as a viable solution to address reverse causality, simultaneous bias, and possible omitted variables in research studies [116, 117]. To ensure the model's accuracy, two standard tests (AR 1 and 2 and the Hansen test) were performed. Following the previous study [118], we control for the potential endogeneity of the independent variables in the TSGMM estimations using two to four periods lags of the same variables as instruments.

We use the following baseline equation to determine the link between risk-taking and bank capital, following the studies of Rahman et al. [3] and Mujtaba et al. [46]:

$$Y_{i,t} = \beta_0 + \beta_1 Y_{i,t-1} + \beta_2 MIV_{i,t} + \beta_q \sum_{q=3}^{6} B_{i,q,t} + \beta_7 I_{i,r,t} + \beta_s \sum_{s=8}^{9} M_{i,s,t} + \varepsilon_{i,t} \qquad (2)$$

In the Eq (2), subscript i and t represent cross-sectional dimensions across banks and time, respectively. Again, q, r, and s represent bank, industry, and macroeconomic-specific perspectives. The vectors B, I, and M consist of bank-specific, industry-specific, and macroeconomic-specific control variables. $\varepsilon_{i,t}$ is an error term.

'$Y_{i,t}$' represents the dependent variable – risk and capital. $Y_{i,t-1}$ denotes the lagged dependent variable. $MIV_{i,t}$ refers to the main independent variables; for the risk equation, capital is the main independent variable, while for the capital equation, risk is the main independent variable. Bank-level control variables presented by $B_{i,q,t}$, including financial intermediation cost, direct taxation, size, and profitability for risk measures, and financial intermediation cost, leverage, size, and profitability for capital measures. $I_{i,r,t}$ present industry-level variable, which includes BSD (Banking Industry Asset to GDP). $M_{i,s,t}$ denotes macroeconomic variables, including GDP growth and inflation.

Eq (2) incorporates risk and capital equations, considering the equation's rank and order condition [119, 120]. However, as our baseline equation employs the same data to estimate risk and capital, the error terms between the two equations may be correlated. Failure to consider simultaneous equation bias when estimating Eq (2) may lead to inconsistent and biased estimators due to the interplay between endogenous variables and random errors. Therefore, it is vital to consider and address this connection to ensure accurate and reliable estimates. Moreover, the concurrent relationship between the error terms ($\varepsilon_{i,t}$) in both risk and capital equation is not by chance; it emerges because both error terms ($\varepsilon_{i,t}$) encapsulate the impact of certain omitted factors that influence the variables under examination. Given the similarities among firms, it is probable that omitted factors exert comparable effects on the risk and capital across different entities. Consequently, the error terms of both risk and capital equations capture analogous effects, resulting in their correlation. To address this issue, the panel GMM approach is an effective system for jointly estimating both equations [107, 121]. Although strict exogeneity and no serial correlations are unrealistic assumptions [119], GMM estimators have demonstrated superior efficiency in panel data compared to fixed effects or random effects estimators, particularly in cases where strict exogeneity assumptions are violated or serial correlation is present [107, 121]. Given the effective control of endogeneity through the

simultaneous equations' framework [107, 119], all estimations in Section 4.1, Section 4.2, and 4.3 employ a TSGMM approach to get empirical findings of Eq (2) to Eq (9). Section 4.3 also uses Two-Stage Least Squares (2SLS) methods, an alternative regression methodology, for robustness checks.

We have extended our baseline equations by adding the non-linear effect of main independent variables on dependent variables, competition, and ownership dummies to analyze the impact of ownership and competition. The extended models are presented as follows:

$$Y_{i,t} = \beta_0 + \beta_1 Y_{i,t-1} + \beta_2 MIV_{i,t} + \beta_3 MIV^2_{i,t} + \beta_q \sum_{q=4}^{7} B_{i,q,t} + \beta_8 I_{i,r,t} + \beta_s \sum_{s=9}^{10} M_{i,s,t} + \varepsilon_{i,t} \quad (3)$$

$$Y_{i,t} = \beta_0 + \beta_1 Y_{i,t-1} + \beta_2 MIV_{i,t} + \beta_3 MIV_{i,t} + \beta_q \sum_{q=4}^{7} B_{i,q,t} + \beta_8 I_{i,r,t} + \beta_s \sum_{s=9}^{10} M_{i,s,t} + \varepsilon_{i,t} \quad (4)$$

$$Y_{i,t} = \beta_0 + \beta_1 Y_{i,t-1} + \beta_2 MIV_{i,t} + \beta_3 MIV_{i,t} \times endo_{i,t} + \beta_q \sum_{q=4}^{7} B_{i,q,t} + \beta_8 I_{i,r,t} + \beta_s \sum_{s=9}^{10} M_{i,s,t}$$
$$+ \varepsilon_{i,t} \quad (5)$$

$$Y_{i,t} = \beta_0 + \beta_1 Y_{i,t-1} + \beta_2 MIV_{i,t} + \beta_3 Ownership_{i,t} + \beta_q \sum_{q=4}^{7} B_{i,q,t} + \beta_8 I_{i,r,t} + \beta_s \sum_{s=9}^{10} M_{i,s,t}$$
$$+ \varepsilon_{i,t} \quad (6)$$

$$Y_{i,t} = \beta_0 + \beta_1 Y_{i,t-1} + \beta_2 MIV_{i,t} + \beta_3 Ownership_{i,t} \times MIV_{i,t} + \beta_q \sum_{q=4}^{7} B_{i,q,t} + \beta_8 I_{i,r,t}$$
$$+ \beta_s \sum_{s=9}^{10} M_{i,s,t} + \varepsilon_{i,t} \quad (7)$$

$$Y_{i,t} = \beta_0 + \beta_1 Y_{i,t-1} + \beta_2 MIV_{i,t} + \beta_3 Ownership_{i,t} \times Comp_{i,t} + \beta_q \sum_{q=4}^{7} B_{i,q,t} + \beta_8 I_{i,r,t}$$
$$+ \beta_s \sum_{s=9}^{10} M_{i,s,t}$$
$$+ \varepsilon_{i,t} \quad (8)$$

$$Y_{i,t} = \beta_0 + \beta_1 Y_{i,t-1} + \beta_2 MIV_{i,t} + \beta_3 Ownership_{i,t} \times Comp_{i,t} \times MIV_{i,t} + \beta_q \sum_{q=4}^{7} B_{i,q,t} + \beta_8 I_{i,r,t}$$
$$+ \beta_s \sum_{s=9}^{10} M_{i,s,t} + \varepsilon_{i,t} \quad (9)$$

Eq (3) shows the nonlinear effect of capital on risk and the nonlinear effect of risk on capital. Eq (4) shows the effects of competition on the dependent variables. Eq (5) demonstrates the joint effects of competition and the main independent variable on the dependent variable. The impact of ownership on the dependent variable is showcased in Eq (6). Eq (7) shows the joint effects of ownership and the main independent variable on the dependent variable. Additionally, Eq (8) presents the joint effects of competition and ownership on the dependent variable. Finally, Eq (9) displays the collective impacts of ownership, competition, and main independent variables on the dependent variable.

Here, Eqs (2) to (9) serve the purpose of testing various hypotheses. Eqs (2) and (3) are tailored to examine hypothesis number (1) and hypothesis (2), respectively. Similarly, Eqs (4) and (5) are composed to scrutinize hypothesis (3), while Eqs (6) and (7) are devised to investigate hypothesis (4). Finally, Eqs (8) and (9) are designed to put hypothesis (5) to the test. In conducting our analysis, we utilized STATA 18 to process and analyze the data collected.

### 3.4 Granger Causality Tests (GCT)

Following previous studies of Nguyen [107], Patel et al. [122], Nguyen and Le [120], Barra and Ruggiero [123], and others, we conduct GCT to examine the possible bidirectional relationship between risk and bank capital. Following the methods of Granger [124], we form the following pairwise Granger causality model:

$$Risk_t = \sum_{j=1}^{m} \alpha_{1j} Risk_{t-j} + \sum_{j=1}^{m} \alpha_{2j} Capital_{t-j} + \varepsilon_{i,t} \qquad (10)$$

$$Capital_t = \sum_{j=1}^{m} \beta_{1j} Capital_{t-j} + \sum_{j=1}^{m} \beta_{2j} Risk_{t-j} + \epsilon_{i,t} \qquad (11)$$

Where, $Risk_t$ represents NPL, $Capital_t$ represents CAR, t represents period (t = 1,2,3,. . . . . ., T), and j signifies the lag lengths. The error terms $\varepsilon_{i,t}$ and $\epsilon_{i,t}$ are treated as two distinct white noise series. It is worth noting that the Granger causality model does not include the extensive controls described in Eq (2) to Eq (9). Significance in the coefficients of the $Risk_{i,t}$ and $Capital_{i,t}$ in the above regressions indicate the presence of the bidirectional relationship or Granger causality between them [124]. For lag selection, we follow the *ad hoc* lag selection process of Jones [125]. As mentioned earlier, our study also addresses endogenous issues, as simultaneous equation model bias may result in inconsistent estimators. However, as Table A1 in Appendix A in S1 Appendix shows, our results suggest that the bidirectional relationship between risk and capital may exist in all cases.

### 3.5 Inflection point

We incorporate the squared term of the leading independent variables in Eq (3) to explore possible non-linear connections. To understand the outcomes of the leading independent variables and their squared terms, we adhere to the method delineated by Hussain and Bashir [126] and Zheng et al. [127] and compute inflection points. An inflection point is a point on a curve at which the curvature changes sign, meaning it is the point where the direction of the curvature changes from concave to convex or vice versa. In the context of our study, identifying the inflection point is essential for understanding the non-linear relationship between bank risk and capital. These points signify instances where the association between variables undergoes a significant transformation, such as a shift in direction from positive to negative or vice versa.

$$\text{Inflection Point} = \frac{-coefficient\ of\ indpendent\ variable}{2 \times coefficient\ of\ the\ squared\ term\ of\ independent\ variable} \qquad (12)$$

Notably, inflection point identification is not feasible when no association changes exist between the variables [128]. Understanding the inflection point is crucial for regulators and bank managers. In the current study, the relationship between risk and capital is a guiding light for setting regulatory capital requirements. This understanding, when applied by bank managers, empowers them to make informed decisions about capital allocation. Ultimately, this leads to better risk management and financial stability in the banking sector.

## 4 Empirical findings and discussions

In this section, we summarize the statistics presented in Table 2 and the results of the multicollinearity test found in Tables 3 and 4. To obtain the empirical results in Tables 5–12, we utilize a TSGMM panel estimator. Additionally, we have conducted the Hansen over-identification test to verify the validity of the instruments used.

**Table 2. Descriptive statistics of the variables.**

| Variable | Obs | Mean | Std. Dev. | Min | Max | Skewness | Kurtosis | Jarque-Bera Test |
|---|---|---|---|---|---|---|---|---|
| CAR | 494 | .149 | .175 | -.284 | 2.207 | 7.397 | 73.899 | 107970.51*** |
| NPL | 494 | .078 | .103 | 0 | .598 | 2.865 | 11.444 | 2143.43*** |
| CFI | 494 | .023 | .014 | -.024 | .104 | -.236 | 5.755 | 160.81*** |
| lever | 494 | .9 | .128 | .047 | 2.493 | 1.243 | 62.319 | 72554.68*** |
| DT | 494 | .446 | .336 | -.801 | 5.486 | 8.319 | 117.811 | 277018.50*** |
| Size | 494 | 11.762 | 1.095 | 6.818 | 14.138 | -1.071 | 5.332 | 206.38*** |
| ROA | 494 | .899 | 1.103 | -7.49 | 6.05 | -1.865 | 17.859 | 4830.97*** |
| BSD | 494 | 48.044 | 2.856 | 41.054 | 51.11 | -1.111 | 3.625 | 109.67*** |
| Boone | 494 | -3.207 | 2.878 | -8.602 | -.082 | -.713 | 2.052 | 60.35*** |
| GDP | 494 | 5.122 | 1.082 | 2.271 | 6.688 | -1.296 | 4.798 | 204.83*** |
| Inflation | 494 | 6.539 | 1.532 | 5.514 | 11.395 | 2.107 | 6.95 | 686.67*** |

The variables are shown in Table 2. The dependent variables, NPL and CAR, have an average value of 7.8% and 14.9%, respectively. The average values for leverage (lever), direct taxation (DT), size, and profitability are 0.9, 0.446, 11.762, and 0.899, respectively. The industry-specific variable, BSD (Banking Sector Development), has a mean value of 48.044. The Boone Indicator (BI), which measures market competition, has an average value of -3.207, which is generally negative. A higher BI value indicates a more intensely competitive market. In this case, the negative value of BI suggests competitiveness among banks in Bangladesh. The mean values for GDP and inflation are 5.122 and 6.359, respectively. Regarding the skewness values, six variables exhibit left skewness, while the remaining variables demonstrate right skewness. Examination of the kurtosis values indicates that only the distribution of Boone displays platykurtic characteristics ($< 3$), while the distributions of other variables are leptokurtic ($>3$). Based on the results of the Jarque–Bera test, it is evident that all variables deviate significantly from a normal distribution at the 1% significance level [129].

Table 3 displays the pairwise correlation between variables, and Table 4 presents the variance inflation factor test results for risk and bank capital. Based on the data, we can conclude that there is no multicollinearity, as no correlations exceed 0.70 for any independent variables. Additionally, the variance inflation factor value is 10, consistent with earlier research (e.g., [34]). Moreover, we conducted panel cointegration tests using the Pedroni and Westerlund

**Table 3. Variance inflation factor.**

| Variable | VIF (Capital Equation) | VIF (Risk Equation) |
|---|---|---|
| CAR | | 1.533 |
| NPL | 1.876 | |
| CFI | 1.804 | 1.38 |
| Boone | 1.799 | 1.782 |
| Lever | 1.384 | |
| DT | | 1.028 |
| Size | 1.515 | 1.671 |
| ROA | 1.714 | 1.363 |
| BSD | 1.465 | 1.466 |
| GDP | 1.481 | 1.472 |
| Inflation | 1.349 | 1.365 |
| Mean VIF | 1.599 | 1.451 |

**Table 4. Matrix of correlations.**

| Variables | (1) | (2) | (3) | (4) | (5) | (6) | (7) | (8) | (9) | (10) | (11) |
|---|---|---|---|---|---|---|---|---|---|---|---|
| (1) CAR | 1.000 | | | | | | | | | | |
| (2) NPL | -0.248 | 1.000 | | | | | | | | | |
| (3) CFI | 0.166 | -0.564 | 1.000 | | | | | | | | |
| (4) Lever | -0.582 | 0.008 | 0.044 | 1.000 | | | | | | | |
| (5) DT | 0.079 | -0.079 | 0.078 | -0.022 | 1.000 | | | | | | |
| (6) Size | -0.568 | 0.108 | -0.270 | 0.470 | 0.006 | 1.000 | | | | | |
| (7) ROA | 0.098 | -0.572 | 0.443 | -0.129 | 0.004 | -0.146 | 1.000 | | | | |
| (8) BSD | 0.089 | -0.009 | 0.020 | -0.076 | 0.047 | -0.173 | -0.028 | 1.000 | | | |
| (9) Boone | -0.045 | -0.069 | -0.071 | -0.044 | 0.020 | 0.013 | 0.068 | -0.345 | 1.000 | | |
| (10) GDP | -0.013 | 0.064 | 0.130 | 0.045 | -0.007 | 0.030 | -0.059 | -0.225 | -0.391 | 1.000 | |
| (11) Inflation | -0.007 | -0.149 | 0.182 | -0.079 | 0.059 | -0.190 | 0.310 | 0.023 | 0.348 | -0.151 | 1.000 |

Note: Pearson's Correlation coefficients.

approaches in line with the methodologies adopted by Batrancea, Rathnaswamy [130], Batrancea, Batrancea [131], and Murtaza, Hongzhong [132]. The results in Tables B1 and B2 in Appendix B in S1 Appendix consistently reject the null hypothesis, indicating cointegration across all panels. Table B1 in S1 Appendix confirms a long-term relationship between the included variables and bank capital, while Table B2 in S1 Appendix validates a similar relationship with bank risk.

## 4.1 Determinants of risk and effect of capital, competition, and ownership

Tables 5 and 6 delve into the connection between risk and capital and how competition and ownership affect that relationship. The regression coefficient of the lag-dependent variable is significant, which means that the effect of the previous year's data carries over to the current year, resulting in a stable risk ratio. Model I of Table 5 shows a negative correlation between capital and risk. This finding implies that more regulatory capital tends to lower credit risk, and well-capitalized banks tend to take fewer risks than their less-capitalized counterparts. This finding is aligned with the findings of Anginer et al. [55].

The study finds that higher financial intermediation costs lead to lower levels of risk-taking among banks from the coefficient of CFI. This finding is aligned with the findings of Gupta et al. [99]. Conversely, direct taxation positively impacts risk, indicating that increased taxation leads to increased financial vulnerability and reduced capital for risk mitigation. Similar findings were also observed in the study of Saeed et al. [100]. The study also finds that a larger asset size can decrease bank risk levels, contradicting the idea of "too big to fail" (see, e.g., [58]).

Additionally, profitability is negatively related to risk, indicating that consistently profitable performance contributes to the overall stability and soundness of the banking sector. This finding is consistent with previous research by Zheng et al. [18]. The industry-level variable BSD has a significant negative association with risk, meaning that developing the banking sector helps banks reduce risks. On the other hand, the macroeconomic variable GDP has a significant positive impact on risk. This indicates that economic growth can lead to increased bank risk as the demand for credit usually increases during economic expansion, exposing banks to take more risks. This result is consistent with the findings of Moudud-Ul-Huq et al. [34]. A negative relationship between inflation and risk suggests that higher inflation levels may lead to reduced risks in the banking sector. This finding is consistent with the findings of Zheng et al. [18].

**Table 5. Determinants of risk and effect of capital.**

| Variable Name | Model I | Model II |
|---|---|---|
| NPL (-1) | 0.8073*** (0.0079) | 0.7964*** (0.0045) |
| CAR | -0.1247*** (0.0066) | -0.2864*** (0.0095) |
| CAR$^2$ | | 0.2334*** (0.0111) |
| Inflection Point | | 0.6136 |
| CFI | -0.5290*** (0.0493) | -0.5174*** (0.0379) |
| DT | 0.0174*** (0.0008) | 0.0169*** (0.0007) |
| Size | -0.0098*** (0.000) | -0.0095*** (0.0004) |
| ROA | -0.0112*** (0.0006) | -0.0068*** (0.0003) |
| BSD | -0.0003* (0.0001) | -0.0002** (0.0001) |
| GDP | 0.0047*** (0.0002) | 0.0046*** (0.0002) |
| Inflation | -0.0010*** (0.0002) | -0.0018*** (0.0001) |
| Constant | 0.1623*** (0.012) | 0.1751*** (0.009) |
| Hansen test (p-value) | 0.205 | 0.126 |
| AR(1) (p-value) | 0.008 | 0.009 |
| AR(2) (p-value) | 0.519 | 0.568 |
| Observations | 450 | 450 |

Notes: The table displays the empirical results obtained from the TSGMM. The dependent variable is the NPL (non-performing loans) to total loans. Standard are errors in parentheses. ***, **, * *denotes the* significance level at the corresponding 1%, 5%, and 10% level. The p-value associated with the Hansen test is called the J statistic. The null hypothesis of the Hansen test posits that there is no correlation between the instruments used and the residuals. Tests for first-order (second-order) correlation using Arellano-Bond orders 1 and 2 are asymptotically N (0,1). In the context of system GMM estimation, these tests assess the first-differenced residuals. The examination of multicollinearity across all models, utilizing the VIF test, consistently revealed VIF values below the threshold of 10, indicating a negligible propensity for multicollinearity. Furthermore, the White test rejected the null hypothesis of heteroscedasticity.

The Model II of Table 5 reveals a significant U-shape relationship between bank capital and risk. To assess the nature of the connection between the variables in the quadratic equation, we determine the inflection point and compare it to the data distribution. The inflection point of the equation is 0.6136 and occurs at approximately the 97th percentile of the capital (CAR) distribution. This result supports hypothesis 1, which suggests a negative relationship between bank capital and risk up to the inflection point; however, the relationship becomes positive once the inflection point is surpassed. These findings are consistent with the findings of Jiang et al. [25]. Our results support both the regulatory hypothesis and the moral hazard hypothesis. While regulatory pressures can encourage prudent risk management by maintaining a certain capital adequacy level, excessive capital may lead to moral hazard behavior, where banks invest in riskier projects to increase shareholder benefits. Our study confirms the optimal capital

**Table 6. Effect of capital, competition, and ownership on risk.**

| Variable Name | Model I | Model II | Model III | Model IV | Model V | Model VI | Model VII | Model VIII | Model IX | Model X |
|---|---|---|---|---|---|---|---|---|---|---|
| NPL (-1) | 0.8026*** (0.0107) | 0.8024*** (0.0083) | 0.7242*** (0.0071) | 0.8048*** (0.0082) | 0.8007*** (0.0079) | 0.8086*** (0.0077) | 0.8038*** (0.0085) | 0.8046*** (0.0084) | 0.8062*** (0.0077) | 0.807*** (0.0079) |
| CAR | -0.1257*** (0.0065) | -0.1638*** (0.0091) | -0.1635*** (0.0064) | -0.1152*** (0.0093) | -0.1269*** (0.0069) | -0.1375*** (0.0061) | -0.1254*** (0.006) | -0.1248*** (0.0061) | -0.1242*** (0.0061) | -0.1241*** (0.0065) |
| CFI | -0.5450*** (0.0572) | -0.5422*** (0.0453) | -0.2517*** (0.0339) | -0.5157*** (0.0459) | -0.4827*** (0.051) | -0.5382*** (0.0487) | -0.4982*** (0.0469) | -0.5053*** (0.0475) | -0.5016*** (0.0488) | -0.5153*** (0.0497) |
| DT | 0.0173*** (0.0008) | 0.0172*** (0.0008) | 0.0142*** (0.0007) | 0.0171*** (0.0008) | 0.0169*** (0.0007) | 0.0178*** (0.0008) | 0.0172*** (0.0008) | 0.0172*** (0.0008) | 0.0172*** (0.0007) | 0.0173*** (0.0008) |
| Size | -0.0097*** (0.0006) | -0.0098*** (0.0006) | -0.0143*** (0.0005) | -0.0098*** (0.0006) | -0.0100*** (0.0005) | -0.0101*** (0.0005) | -0.0093*** (0.0006) | -0.0094*** (0.0006) | -0.0093*** (0.0006) | -0.0096*** (0.0006) |
| ROA | -0.0115*** (0.0005) | -0.0113*** (0.0003) | -0.0109*** (0.0004) | -0.0114*** (0.0003) | -0.0114*** (0.0004) | -0.0108*** (0.0004) | -0.0116*** (0.0005) | -0.0115*** (0.0005) | -0.0114*** (0.0004) | -0.0113*** (0.0004) |
| BSD | -0.0006*** (0.0002) | -0.0008*** (0.0001) | -0.0007*** (0.0001) | -0.0002* (0.0001) | 0.0001 (0.0001) | -0.0002* (0.0001) | -0.0002* (0.0001) | -0.0002* (0.0001) | -0.0002 (0.0001) | -0.0002 (0.0001) |
| Boone | -0.0006** (0.0002) | | | | | | | | | |
| Boone×CAR | | -0.0087*** (0.0006) | | | | | | | | |
| POD | | | -0.0460*** (0.0013) | | | | | | | |
| POD×CAR | | | | -0.01321* (0.0108) | | | | | | |
| POD×Boone | | | | | -0.0011*** (0.0002) | | | | | |
| POD×Boone×CAR | | | | | | -0.0013*** (0.0005) | | | | |
| IOD | | | | | | | -0.00374** (0.0018) | | | |
| IOD×CAR | | | | | | | | -0.0198** (0.0099) | | |
| IOD×Boone | | | | | | | | | -0.0007** (0.0003) | |
| IOD×Boone×CAR | | | | | | | | | | -0.0047* (0.0025) |
| GDP | 0.0041 (0.0003) | 0.0034*** (0.0002) | 0.0032*** (0.0002) | 0.0048*** (0.0001) | 0.0059*** (0.0003) | 0.0044*** (0.0002) | 0.0048*** (0.0002) | 0.0048*** (0.0002) | 0.0049*** (0.0002) | 0.0048*** (0.0002) |
| Inflation | -0.0006*** (0.0002) | -0.00048** (0.0002) | -0.0026*** (0.0002) | -0.0010*** (0.0002) | -0.0016*** (0.0002) | -0.0008*** (0.0002) | -0.0010*** (0.0002) | -0.0010*** (0.0002) | -0.0010*** (0.0002) | -0.001*** (0.0002) |
| Constant | 0.1766*** (0.0163) | 0.1930*** (0.014) | 0.3025*** (0.0097) | 0.1610*** (0.0122) | 0.1435*** (0.0115) | 0.1661*** (0.0128) | 0.1556 (0.0127) | 0.1569*** (0.0127) | 0.1511 (0.0138) | 0.1552*** (0.0138) |
| Hansen test (p-value) | 0.204 | 0.210 | 0.246 | 0.223 | 0.201 | 0.171 | 0.228 | 0.227 | 0.231 | 0.112 |
| AR(1) (p-value) | 0.007 | 0.007 | 0.002 | 0.008 | 0.009 | 0.008 | 0.008 | 0.008 | 0.008 | 0.008 |
| AR(2) (p-value) | 0.603 | 0.523 | 0.482 | 0.539 | 0.565 | 0.488 | 0.540 | 0.638 | 0.539 | 0.530 |
| Observations | 450 | 450 | 450 | 450 | 450 | 450 | 450 | 450 | 450 | 450 |

Notes: The table displays the empirical results obtained from the TSGMM. The dependent variable is the NPL (non-performing loans) to total loans. Standard errors are in parentheses. ***, **, * *denotes the* significance level at the corresponding 1%, 5%, and 10% level. The p-value associated with the Hansen test is called the J statistic. The null hypothesis of the Hansen test posits that there is no correlation between the instruments used and the residuals. Tests for first-order (second-order) correlation using Arellano-Bond orders 1 and 2 are asymptotically N (0,1). In the context of system GMM estimation, these tests assess the first-differenced residuals. The examination of multicollinearity across all models, utilizing the VIF test, consistently revealed VIF values below the threshold of 10, indicating a negligible propensity for multicollinearity. Furthermore, the White test rejected the null hypothesis of heteroscedasticity.

**Table 7. Determinants of capital and effect of risk.**

| Variable Name | Model I | Model II |
|---|---|---|
| CAR (-1) | 0.4392*** (0.0022) | 0.4471*** (0.0018) |
| NPL | -0.0499*** (0.0062) | -0.0918*** (0.0115) |
| NPL$^2$ | | -0.2899*** (0.0162) |
| CFI | 0.17557*** (0.0297) | 0.24*** (0.0328) |
| Lever | -0.1233*** (0.012) | -0.1173*** (0.0121) |
| Size | -0.0021*** (0.0006) | -0.0031*** (0.0006) |
| ROA | 0.0176*** (0.0004) | 0.018*** (0.0004) |
| BSD | -0.0021*** (0.0001) | -0.0023*** (0.0001) |
| GDP | -0.0025*** (0.0002) | -0.0028*** (0.0002) |
| Inflation | -0.0049*** (0.0002) | -0.0047*** (0.0002) |
| Constant | 0.3437*** (0.0111) | 0.3488*** (0.0101) |
| Hansen Test (P-value) | 0.230 | 0.173 |
| AR (1) (P-value) | 0.087 | 0.090 |
| AR (2) (P-value) | 0.383 | 0.354 |
| Observations | 450 | 450 |

Notes: The table displays the empirical results obtained from the TSGMM. The dependent variable is the CAR (Capital Adequacy Ratio). Standard errors are in parentheses. ***, **, * *denotes the* significance level at the corresponding 1%, 5%, and 10% level. The p-value associated with the Hansen test is called the J statistic. The null hypothesis of the Hansen test posits that there is no correlation between the instruments used and the residuals. Tests for first-order (second-order) correlation using Arellano-Bond orders 1 and 2 are asymptotically N (0,1). In the context of system GMM estimation, these tests assess the first-differenced residuals. The examination of multicollinearity across all models, utilizing the VIF test, consistently revealed VIF values below the threshold of 10, indicating a negligible propensity for multicollinearity. Furthermore, the White test rejected the null hypothesis of heteroscedasticity.

structure theorem, stressing the importance of maintaining a balanced capital structure for effective risk management.

The empirical findings of Eq (3) – Eq (9) related to the risk equation presented in the Model (I) to Model (X) of Table 6. To establish direct proportionality to competition, we use the opposite of the BI (i.e., $-\beta_t$) by following the study of Tabak et al. [93]. The correlation between market competition measured by the opposite Boone indicator and risk is negative. Our results advocate that a higher degree of competition leads to a decrease in risk-taking in Bangladeshi commercial banks [35]. This finding supports the competition stability hypothesis (e.g., [33]). The interim term between competition and capital shows a significant negative link with risk. This finding supports hypothesis 2 and posits that in a competitive market, the holdings of more regulatory capital can decrease the banks' credit risk.

From the ownership scene, private-owned and Islamic commercial banks are inversely connected with credit risk-taking, meaning that the presence of private-owned (state-owned) and

**Table 8. Effect of risk, competition, and ownership on capital.**

| Variable Name | Model I | Model II | Model III | Model IV | Model V | Model VI | Model VII | Model VIII | Model IX | Model X |
|---|---|---|---|---|---|---|---|---|---|---|
| CAR (-1) | 0.4407*** (0.0022) | 0.4390*** (0.0025) | 0.4267*** (0.0022) | 0.4284*** (0.0028) | 0.4381*** (0.0024) | 0.441*** (0.0023) | 0.439*** (0.0017) | 0.4385*** (0.0022) | 0.4396*** (0.0026) | 0.4382*** (0.002) |
| NPL | -0.04922*** (0.0065) | -0.1270*** (0.0063) | -0.1067*** (0.0048) | -0.0322*** (0.0063) | -0.0541*** (0.007) | -0.0561*** (0.0083) | -0.0485*** (0.0065) | -0.0481*** (0.0063) | -0.0596*** (0.007) | -0.0467*** (0.0066) |
| CFI | 0.1693*** (0.0338) | 0.0742* (0.0417) | 0.3118*** (0.0298) | 0.2553*** (0.0309) | 0.2092*** (0.03) | 0.1123*** (0.0336) | 0.2496*** (0.0278) | 0.1859*** (0.0295) | 0.4572*** (0.0198) | 0.1886*** (0.0301) |
| Lever | -0.1217*** (0.0123) | -0.1222*** (0.0134) | -0.1124*** (0.0106) | -0.1177*** (0.0144) | -0.1213*** (0.0117) | -0.1295*** (0.0109) | -0.1264*** (0.0104) | -0.1289*** (0.0127) | -0.1186*** (0.0171) | -0.1266*** (0.0118) |
| Size | -0.0021*** (0.0006) | -0.0024*** (0.0006) | -0.0057*** (0.0005) | -0.0043*** (0.0006) | -0.0023*** (0.0006) | -0.0012* (0.0007) | -0.0011** (0.0005) | -0.0019*** (0.0005) | -0.0015** (0.0007) | -0.0021*** (0.0006) |
| ROA | 0.0175*** (0.0003) | 0.0172*** (0.0004) | 0.0167*** (0.0003) | 0.0164*** (0.0003) | 0.0175*** (0.0004) | 0.018*** (0.0004) | 0.0174*** (0.0004) | 0.0176*** (0.0003) | 0.0164*** (0.0003) | 0.0176*** (0.0004) |
| BSD | -0.0025*** (0.00008) | -0.0028*** (0.00009) | -0.0024*** (0.0001) | -0.0023*** (0.00009) | -0.0018*** (0.0001) | -0.0024*** (0.0001) | -0.0021*** (0.00008) | -0.0021*** (0.0001) | -0.0013*** (0.0001) | -0.0021*** (0.0001) |
| Boone | 0.0008*** (0.00006) | | | | | | | | | |
| Boone×NPL | | -0.0162*** (0.0004) | | | | | | | | |
| POD | | | -0.0259*** (0.0017) | | | | | | | |
| POD×NPL | | | | 0.1189*** (0.0113) | | | | | | |
| POD×Boone | | | | | 0.0007*** (0.0001) | | | | | |
| POD×Boone×NPL | | | | | | 0.0137*** (0.0021) | | | | |
| IOD | | | | | | | -0.0059** (0.0025) | | | |
| IOD×NPL | | | | | | | | -0.0534 (0.0415) | | |
| IOD×Boone | | | | | | | | | -0.0124*** (0.0011) | |
| IOD×Boone×NPL | | | | | | | | | | -0.0119* (0.0067) |
| GDP | -0.0035*** (0.0002) | -0.0040*** (0.0001) | -0.0028*** (0.0001) | -0.0027*** (0.0001) | -0.0016*** (0.0002) | -0.0033*** (0.0002) | -0.0026*** (0.0001) | -0.0021*** (0.0002) | -0.0011*** (0.0002) | -0.0026*** (0.0002) |
| Inflation | -0.0044*** (0.0002) | -0.0044*** (0.0002) | -0.0060*** (0.0003) | -0.0057*** (0.0002) | -0.0055*** (0.0002) | -0.0044*** (0.0001) | -0.0049*** (0.0002) | -0.0049*** (0.0002) | -0.0058 (0.0002) | -0.0049*** (0.0002) |
| Constant | 0.3591*** (0.0083) | 0.3860*** (0.0108) | 0.4249*** (0.0143) | 0.3851*** (0.0125) | 0.3291*** (0.0125) | 0.3507*** (0.0097) | 0.3327*** (0.0086) | 0.3443*** (0.0084) | 0.2521*** (0.0133) | 0.3472*** (0.01) |
| Hansen test (p-value) | 0.254 | 0.221 | 0.233 | 0.246 | 0.232 | 0.203 | 0.220 | 0.148 | 0.148 | 0.135 |
| AR(1) (p-value) | 0.088 | 0.092 | 0.083 | 0.082 | 0.084 | 0.087 | 0.083 | 0.081 | 0.066 | 0.084 |
| AR(2) (p-value) | 0.331 | 0.270 | 0.381 | 0.349 | 0.439 | 0.342 | 0.384 | 0.389 | 0.312 | 0.0376 |
| Observations | 450 | 450 | 450 | 450 | 450 | 450 | 450 | 450 | 450 | 450 |

Notes: The table displays the empirical results obtained from the TSGMM. The dependent variable is the CAR (Capital Adequacy Ratio). Standard errors are in parentheses. ***, **, * *denotes the* significance level at the corresponding 1%, 5%, and 10% level. The p-value associated with the Hansen test is called the J statistic. The null hypothesis of the Hansen test posits that there is no correlation between the instruments used and the residuals. Tests for first-order (second-order) correlation using Arellano-Bond orders 1 and 2 are asymptotically N (0,1). In the context of system GMM estimation, these tests assess the first-differenced residuals. The examination of multicollinearity across all models, utilizing the VIF test, consistently revealed VIF values below the threshold of 10, indicating a negligible propensity for multicollinearity. Furthermore, the White test rejected the null hypothesis of heteroscedasticity.

**Table 9. Determinants of risk and effect of capital.**

| LLP (-1) | Model I | Model II |
|---|---|---|
| LLP (-1) | 0.7214*** | 0.7082*** |
| | (0.0038) | (0.0026) |
| T1C | -0.0792*** | -0.1724*** |
| | (0.0056) | (0.0067) |
| $T1C^2$ | | 0.318*** |
| | | (0.017) |
| Inflection Point | | 0.271 |
| CFI | -0.0741*** | -0.0837*** |
| | (0.0112) | (0.0114) |
| DT | 0.0012*** | 0.0013*** |
| | (0.0002) | (0.0001) |
| Size | -0.0027*** | -0.0021*** |
| | (0.0002) | (0.0002) |
| ROA | -0.0087*** | -0.0068*** |
| | (0.0002) | (0.0002) |
| BSD | -0.0004*** | -0.0005*** |
| | (0.00004) | (0.00003) |
| GDP | 0.0010*** | 0.0011*** |
| | (0.00009) | (0.00008) |
| Inflation | -0.0004* | -0.0003*** |
| | (0.0001) | (0.0001) |
| Constant | 0.0280*** | 0.0227*** |
| | (0.0042) | (0.005) |
| Hansen test (p-value) | 0.234 | 0.147 |
| AR(1) (p-value) | 0.028 | 0.031 |
| AR(2) (p-value) | 0.351 | 0.318 |
| Observations | 450 | 450 |

Notes: The table displays the empirical results obtained from the TSGMM. The dependent variable is the LLP (Loan Loss provision to Total Loan) ratio. Standard errors are in parentheses. ***, **, * *denotes the* significance level at the corresponding 1%, 5%, and 10% level. The p-value associated with the Hansen test is called the J statistic. The null hypothesis of the Hansen test posits that there is no correlation between the instruments used and the residuals. Tests for first-order (second-order) correlation using Arellano-Bond orders 1 and 2 are asymptotically N (0,1). In the context of system GMM estimation, these tests assess the first-differenced residuals. The examination of multicollinearity across all models, utilizing the VIF test, consistently revealed VIF values below the threshold of 10, indicating a negligible propensity for multicollinearity. Furthermore, the White test rejected the null hypothesis of heteroscedasticity.

Islamic-owned (Conventional) commercial banks causes decreasing (increasing) credit risk. This also implies that private and Islamic banks are more risk-averse than state-owned and conventional commercial banks (see, e.g., [18]). Since state-owned banks directly liaise with the government, they are more likely to take more credit risks [18, 86]. This finding aligns with the "political view of state-government ownership," which states that state-owned banks provide political sponsorship, and their decisions are influenced by the country's political activities and elections [133].

However, the interim term between private (state) and Islamic (conventional) ownership dummies and capital demonstrates a significant negative (positive) relationship, revealing that in private-owned and Islamic commercial banks, well (low) capitalized banks are likely to take fewer (more) risks, whereas in state-owned and conventional commercial banks well-capitalized banks generally take more risks. These findings support hypothesis 3.

**Table 10. Effect of capital, competition, and ownership on risk.**

| Variable Name | Model I | Model II | Model III | Model IV | Model V | Model VI | Model VII | Model VIII | Model IX | Model X |
|---|---|---|---|---|---|---|---|---|---|---|
| LLP (-1) | 0.7233*** (0.0035) | 0.7207*** (0.004) | 0.6331*** (0.0032) | 0.7108*** (0.0041) | 0.7035*** (0.0037) | 0.7136*** (0.036) | 0.7152*** (0.0033) | 0.7156*** (0.0034) | 0.7165*** (0.0032) | 0.7185*** (0.0037) |
| T1C | -0.077*** (0.0055) | -0.0686*** (0.0062) | -0.1153*** (0.0051) | -0.0672*** (0.006) | -0.086*** (0.0046) | -0.0419*** (0.0052) | -0.0878*** (0.0047) | -0.0875*** (0.0048) | -0.0862*** (0.0047) | -0.081*** (0.0054) |
| CFI | -0.0544*** (0.0148) | -0.0658*** (0.0117) | -0.1373*** (0.0158) | -0.0654*** (0.0116) | -0.0386*** (0.013) | -0.054*** (0.0119) | -0.0626*** (0.0081) | -0.0596*** (0.0078) | -0.0793*** (0.0075) | -0.0454*** (0.0139) |
| DT | 0.0012*** (0.0002) | 0.0013*** (0.0002) | -0.0008*** (0.0002) | 0.001*** (0.0002) | 0.0006*** (0.0002) | 0.0009*** (0.0002) | 0.0011*** (0.0001) | 0.0011*** (0.0001) | 0.0012*** (0.0002) | 0.0008* (0.0004) |
| Size | -0.0026*** (0.0002) | -0.0026*** (0.0002) | -0.0051*** (0.0002) | -0.0027*** (0.0002) | -0.0029*** (0.0002) | -0.0026*** (0.0002) | -0.0027*** (0.0002) | -0.0027*** (0.0002) | -0.0028*** (0.0002) | -0.0024*** (0.0003) |
| ROA | -0.0088*** (0.0002) | -0.0088*** (0.0001) | -0.0078*** (0.0002) | -0.0088*** (0.0002) | -0.0085*** (0.0001) | -0.009*** (0.0001) | -0.0088*** (0.0001) | -0.0088*** (0.0001) | -0.0088*** (0.0001) | -0.0087*** (0.0002) |
| BSD | -0.0007*** (0.00004) | -0.0004*** (0.00004) | -0.00008** (0.00004) | -0.0004*** (0.00005) | 0.0008*** (0.00004) | 0.0007*** (0.00003) | -0.0004*** (0.00004) | -0.0004*** (0.00004) | -0.0005*** (0.00004) | 0.0004*** (0.00006) |
| Boone | -0.0005*** (0.00002) | | | | | | | | | |
| Boone×T1C | | -0.002*** (0.0005) | | | | | | | | |
| POD | | | -0.0249*** (0.0008) | | | | | | | |
| POD×T1C | | | | -0.0159*** (0.0045) | | | | | | |
| POD×Boone | | | | | -0.001*** (0.00005) | | | | | |
| POD×Boone×T1C | | | | | | -0.0097*** (0.0007) | | | | |
| IOD | | | | | | | -0.0034*** (0.001) | | | |
| IOD×T1C | | | | | | | | -0.055*** (0.0149) | | |
| IOD×Boone | | | | | | | | | -0.0004*** (0.0001) | |
| IOD×Boone×T1C | | | | | | | | | | -0.0078*** (0.0024) |
| GDP | 0.0017*** (0.0001) | 0.0012*** (0.00008) | 0.0001** (0.00004) | 0.001*** (0.0001) | 0.0021*** (0.00009) | 0.0017*** (0.00007) | 0.0011*** (0.00007) | 0.0011*** (0.00008) | 0.0012*** (0.00006) | 0.0009*** (0.0001) |
| Inflation | -0.00006 (0.0001) | -0.0003*** (0.0001) | -0.0003*** (0.0001) | -0.0004*** (0.0001) | -0.0003*** (0.0001) | -0.00002 (0.0001) | -0.0004*** (0.00009) | -0.0005*** (0.00009) | -0.0004*** (0.0001) | -0.0004*** (0.0001) |
| Constant | 0.0135*** (0.0042) | 0.0227*** (0.0045) | 0.1055*** (0.0052) | 0.0271*** (0.0046) | 0.0121*** (0.0039) | 0.0114*** (0.004) | 0.0276*** (0.004) | 0.0271*** (0.0041) | 0.0247*** (0.0043) | 0.024*** (0.0044) |
| Hansen test (p-value) | 0.223 | 0.201 | 0.215 | 0.249 | 0.222 | 0.169 | 0.245 | 0.238 | 0.198 | 0.102 |
| AR(1) (p-value) | 0.039 | 0.030 | 0.046 | 0.039 | 0.040 | 0.057 | 0.037 | 0.032 | 0.044 | 0.051 |
| AR(2) (p-value) | 0.357 | 0.357 | 0.357 | 0.355 | 0.364 | 0.375 | 0.245 | 0.351 | 0.351 | 0.353 |
| Observations | 450 | 450 | 450 | 450 | 450 | 450 | 450 | 450 | 450 | 450 |

Notes: The table displays the empirical results obtained from the TSGMM. The dependent variable is the LLP (Loan Loss provision to Total Loan) ratio. Standard errors are in parentheses. ***, **, * *denotes the* significance level at the corresponding 1%, 5%, and 10% level. The p-value associated with the Hansen test is called the J statistic. The null hypothesis of the Hansen test posits that there is no correlation between the instruments used and the residuals. Tests for first-order (second-order) correlation using Arellano-Bond orders 1 and 2 are asymptotically N (0,1). In the context of system GMM estimation, these tests assess the first-differenced residuals. The examination of multicollinearity across all models, utilizing the VIF test, consistently revealed VIF values below the threshold of 10, indicating a negligible propensity for multicollinearity. Furthermore, the White test rejected the null hypothesis of heteroscedasticity.

**Table 11. Determinants of capital and effect of risk.**

| Variable Name | Model I | Model II |
|---|---|---|
| T1C (-1) | 0.5014*** | 0.4903*** |
| | (0.0124) | (0.0115) |
| LLP | -0.0942*** | -0.1773*** |
| | (0.0161) | (0.0277) |
| $LLP^2$ | | -1.0214*** |
| | | (0.0469) |
| CFI | 0.0412** | 0.1849*** |
| | (0.0166) | (0.02) |
| Lever | -0.0977*** | -0.0984*** |
| | (0.0221) | (0.0226) |
| Size | -0.0031*** | -0.0047*** |
| | (0.0002) | (0.0003) |
| ROA | 0.0105*** | 0.0102*** |
| | (0.0006) | (0.0006) |
| BSD | -0.0005*** | -0.0004*** |
| | (0.00007) | (0.00009) |
| GDP | -0.0006*** | -0.0008*** |
| | (0.00008) | (0.0001) |
| Inflation | -0.0017*** | -0.0019*** |
| | (0.0001) | (0.0001) |
| Constant | 0.1937*** | 0.2036*** |
| | (0.0225) | (0.0193) |
| Hansen Test (P-value) | 0.218 | 0.200 |
| AR (1) (P-value) | 0.028 | 0.036 |
| AR (2) (P-value) | 0.437 | 0.475 |
| Observations | 450 | 450 |

Notes: The table displays the empirical results obtained from the TSGMM. The dependent variable is the T1C (Tier1 Capital) ratio. Standard errors are in parentheses. ***, **, * *denotes the* significance level at the corresponding 1%, 5%, and 10% level. The p-value associated with the Hansen test is called the J statistic. The null hypothesis of the Hansen test posits that there is no correlation between the instruments used and the residuals. Tests for first-order (second-order) correlation using Arellano-Bond orders 1 and 2 are asymptotically N (0,1). In the context of system GMM estimation, these tests assess the first-differenced residuals. The examination of multicollinearity across all models, utilizing the VIF test, consistently revealed VIF values below the threshold of 10, indicating a negligible propensity for multicollinearity. Furthermore, the White test rejected the null hypothesis of heteroscedasticity.

The interaction between competition and private-owned (state-owned) banks has a significant negative (positive) effect on the risk. The interaction between competition and Islamic (conventional) banks also significantly negatively (positively) affects the risk. The findings advocate that with increased market competition, private-owned and Islamic commercial banks are likely to take fewer credit risks. In contrast, state-owned and conventional banks tend to take high risks. These findings conform to the findings of Zheng et al. [18]. Again, the interaction among competition, capital, and private-owned (State-owned) and Islamic (conventional) banks have significant adverse (positive) effects on bank risk. This suggests that private and Islamic banks with strong capital positions in the competitive market can manage risk more effectively than state-owned and conventional banks. These findings indicate that ownership and competition significantly affect the relationship between capital and bank risk and support hypothesis 4.

**Table 12. Effect of risk, competition, and ownership on capital.**

| Variable Name | Model I | Model II | Model III | Model IV | Model V | Model VI | Model VII | Model VIII | Model IX | Model X |
|---|---|---|---|---|---|---|---|---|---|---|
| T1C (-1) | 0.5003*** (0.0128) | 0.4905*** (0.0104) | 0.483*** (0.0072) | 0.4746*** (0.0098) | 0.4878*** (0.0111) | 0.4921*** (0.0105) | 0.4918*** (0.0084) | 0.4887*** (0.0067) | 0.4952*** (0.0078) | 0.4936*** (0.0059) |
| LLP | -0.089*** (0.0177) | -0.1825*** (0.0249) | -0.2259*** (0.0116) | -0.1354*** (0.0135) | -0.1044*** (0.0127) | -0.1021*** (0.0135) | -0.0256* (0.0139) | -0.0259*** (0.0088) | -0.0194** (0.0093) | -0.0201*** (0.006) |
| CFI | 0.0473*** (0.0163) | 0.0547*** (0.0188) | 0.1598*** (0.0272) | 0.0813*** (0.019) | 0.103*** (0.0135) | 0.1131*** (0.0157) | 0.1651** (0.0319) | 0.1453*** (0.0228) | 0.1482*** (0.0203) | 0.1417*** (0.0249) |
| Lever | -0.0952*** (0.0228) | -0.1096*** (0.0191) | -0.0887*** (0.0128) | -0.1109*** (0.0188) | -0.1142*** (0.0207) | -0.1081*** (0.0194) | -0.1068*** (0.0172) | -0.1123*** (0.013) | -0.0988*** (0.0153) | -0.1035*** (0.0104) |
| Size | -0.0033*** (0.0002) | -0.0033*** (0.0002) | -0.0055*** (0.0004) | -0.0043*** (0.0003) | -0.0034*** (0.0002) | -0.0032*** (0.0001) | -0.0022*** (0.0001) | -0.0026*** (0.0003) | -0.0027*** (0.0002) | -0.0025*** (0.0001) |
| ROA | 0.0107*** (0.0007) | 0.01*** (0.0005) | 0.009*** (0.0004) | 0.0084*** (0.0006) | 0.01*** (0.0005) | 0.0095*** (0.0005) | 0.0118*** (0.0004) | 0.0118*** (0.0004) | 0.0122*** (0.0004) | 0.012*** (0.0003) |
| BSD | -0.0007*** (0.00007) | -0.0009*** (0.00009) | -0.0006*** (0.0001) | -0.0006*** (0.0001) | -0.0003*** (0.00009) | -0.0003*** (0.0001) | -0.0006*** (0.00009) | -0.0007*** (0.00007) | -0.0006*** (0.00009) | -0.0006*** (0.00005) |
| Boone | 0.0003*** (0.00004) | | | | | | | | | |
| Boone×LLP | | -0.0226*** (0.0022) | | | | | | | | |
| POD | | | -0.0196*** (0.002) | | | | | | | |
| POD×LLP | | | | 0.206*** (0.0259) | | | | | | |
| POD×Boone | | | | | 0.0006*** (0.00004) | | | | | |
| POD×Boone×LLP | | | | | | 0.0267*** (0.0024) | | | | |
| IOD | | | | | | | -0.0047*** (0.0011) | | | |
| IOD×LLP | | | | | | | | -0.0331 (0.0294) | | |
| IOD×Boone | | | | | | | | | -0.0005*** (0.0001) | |
| IOD×Boone×LLP | | | | | | | | | | -0.0121** (0.0052) |
| GDP | -0.0012*** (0.00009) | -0.0015*** (0.0001) | -0.0009*** (0.0001) | -0.0008*** (0.0001) | -0.0002 (0.0001) | -0.0002** (0.0001) | -0.0011*** (0.0001) | -0.001*** (0.0001) | -0.001*** (0.00007) | -0.001*** (0.00008) |
| Inflation | -0.0015*** (0.0001) | -0.0013*** (0.0001) | -0.0023*** (0.0001) | -0.0021 (0.0001) | -0.0022*** (0.0001) | -0.002*** (0.0001) | -0.0018*** (0.0001) | -0.0018*** (0.0001) | -0.0019*** (0.0001) | -0.0019*** (0.0001) |
| Constant | 0.2028*** (0.0227) | 0.2306*** (0.0183) | 0.2447*** (0.0163) | 0.2379*** (0.0175) | 0.2037*** (0.0209) | 0.1967*** (0.0214) | 0.1967*** (0.02) | 0.2076*** (0.0147) | 0.1917*** (0.0176) | 0.1948*** (0.0116) |
| Hansen test (p-value) | 0.299 | 0.322 | 0.285 | 0.278 | 0.210 | 0.214 | 0.294 | 0.245 | 0.234 | 0.286 |
| AR(1) (p-value) | 0.026 | 0.036 | 0.025 | 0.032 | 0.030 | 0.025 | 0.027 | 0.033 | 0.023 | 0.022 |
| AR(2) (p-value) | 0.472 | 0.532 | 0.419 | 0.430 | 0.403 | 0.373 | 0.464 | 0.438 | 0.468 | 0.465 |
| Observations | 450 | 450 | 450 | 450 | 450 | 450 | 450 | 450 | 450 | 450 |

Notes: The table displays the empirical results obtained from the TSGMM. The dependent variable is the T1C (Tier1 Capital) ratio. Standard errors are in parentheses. ***, **, * *denotes the* significance level at the corresponding 1%, 5%, and 10% level. The p-value associated with the Hansen test is called the J statistic. The null hypothesis of the Hansen test posits that there is no correlation between the instruments used and the residuals. Tests for first-order (second-order) correlation using Arellano-Bond orders 1 and 2 are asymptotically N (0,1). In the context of system GMM estimation, these tests assess the first-differenced residuals. The examination of multicollinearity across all models, utilizing the VIF test, consistently revealed VIF values below the threshold of 10, indicating a negligible propensity for multicollinearity. Furthermore, the White test rejected the null hypothesis of heteroscedasticity.

## 4.2 Determinants of capital and effect of risk, competition, and ownership

Tables 7 and 8 show the connection between capital and risk and how competition and ownership affect that relationship. The regression coefficient of the lag-dependent variable is significant, which means that the effect of the previous year's data carries over to the current year, resulting in a stable risk ratio. Model I of Table 7 shows a significant negative relationship between bank risk and capital. The risk coefficient implies that an increase in bank risk is associated with a decline in the CAR ratio. This finding is aligned with the findings of Anh [48].

The study finds that the association between financial intermediation cost and capital is significant and positive, meaning that meeting the expenses of capital banks increases their financial intermediation costs. This is aligned with Rahman et al. [11] findings. The study also finds a significant positive correlation between leverage and capital, indicating that banks effectively use debt to enhance their capital, which conforms to the discoveries of Zheng et al. [18]. On the other hand, the study highlights an inverse relationship between bank size and capital, suggesting that larger banks may struggle to maintain higher capital ratios due to allocating more capital to manage higher-risk-weighted assets. This finding is akin to the conclusions of Saif-Alyousfi and Saha [35].

Furthermore, the study finds a positive correlation between profitability and capital, implying that more profitable banks can boost their capital base by retaining earnings, which aligns with the findings of Garel et al. [49]. The industry-level variable BSD has a significant negative relationship with capital, meaning that developing the banking sector minimizes the banks' capital requirements. Moreover, the macroeconomic variable GDP has an adverse effect on capital, as banks may engage in risky lending practices to maximize returns during economic expansion, leading to a decrease in bank capital. This result is incongruent with the findings of Moudud-Ul-Huq et al. [17]. A negative relationship between inflation and risk confirms the findings of Alqahtani et al. [94].

Model II of Table 7 reveals a significant negative relationship between bank risk and capital. Moreover, we also see a significant negative correlation between the squared term of risk and capital, indicating a notable nonlinear negative impact of bank capital on bank risk. This finding supports hypothesis 1 and suggests that banks with higher levels of risk may find it challenging to maintain their capital adequacy, which could result in difficulties in maintaining adequate capital buffers. Unlike in section 4.1, determining the inflection point in this case is not feasible since it would require a reversal in the direction of their correlation [128], making it impossible to measure. This finding also suggests that the nonlinear negative effects of banks' risk on bank capital may indicate agency problems (see, e.g., [12]. This is because management could engage in risky behavior against shareholders' interests, posing a risk to the bank's capital adequacy.

The empirical findings of Eq (3) – Eq (9) related to the risk equation presented in Model (I) to Model (X) of Table 8. To establish direct proportionality to competition, we use the opposite of the BI (i.e., $-\beta_t$) by following the study of Tabak et al. [93]. The relationship between market competition measured by the opposite Boone indicator and capital is positive. Our findings suggest that in the highly competitive market, banks generally tend to hold more regulatory capital. The interim term between Boone indicator and risk reveals a negative association with capital. This means that banks' capital decreases in a low degree of a competitive market with increased risks, and undercapitalized banks take more risks, and vice versa. This finding confirms hypothesis 2 and supports the charter value view of competition (e.g., [8, 33]).

From the ownership context, the coefficient value of private-owned and Islamic commercial banks is negatively related to capital, meaning that the presence of state-owned and conventional commercial banks ensures more regulatory capital in the market. In contrast,

privately owned and Islamic commercial banks tend to hold less regulatory capital. The coefficient value of credit risk with the private ownership dummy depicts a significant positive alliance with the capital. Again, the coefficient value of credit risk with the Islamic ownership dummy depicts a significant negative association with the capital. These findings state that with high risk-taking, private-owned (state-owned) and conventional (Islamic) commercial banks generally hold more (less) capital. It also implies that well-capitalized private-owned and conventional (state-owned and Islamic) commercial banks take more (less) risks and vice versa. These findings support the hypothesis 3.

The interaction between private (state) ownership dummy and competition exhibits a significantly positive (negative) association with capital. Again, the interaction between Islamic (conventional) ownership dummies and competition exhibits a significantly negative (positive) association with capital. This finding implies that privately owned and conventional commercial banks hold more capital in the competitive market. On the other hand, the capital of state-owned and Islamic commercial banks decreases. For state-owned banks, this may be due to political sponsorship and political pressure they tend to take on risky projects. These findings align with the "political view of state-government ownership [133].

Again, the interaction term among competition, risk, and private-owned (State-owned) banks have significant positive (negative) effects on capital. Again, the interaction among competition, risk, and Islamic (conventional) private banks have significant negative (positive) effects on bank capital. This indicates that as the level of risk increases, the joint effects of competition and state (private) and Islamic (conventional) ownership become more pronounced in negative (positive) impact on bank capital. These suggest that state-owned and Islamic banks in competitive environments facing higher risks tend to experience a reduction in their capital base. It also indicates that managing risk in competitive environments presents more challenges for state-owned and Islamic banks in maintaining higher capital ratios than private and conventional commercial banks. These findings highlight the significant impact of ownership and competition on the relationship between risk and capital, supporting hypothesis 4.

### 4.3 Robustness checks

To reinforce the resilience of the model, we conduct robustness checks by following previous empirical investigations, including Berger et al. [90], Liu et al. [37], Rahman et al. [11], Zheng et al. [67], and Belkhaoui et al. [134] by utilizing different specifications of the dependent and main independent variables. We use the LLP ratio as an alternative proxy for bank risk-taking. Furthermore, we use the Tier 1 Capital ratio (T1C), the Tier 1 capital to total assets ratio, as an alternative proxy of bank capital. We also consider all independent variables in the models as instrument variables except endogenous variables. However, the instruments are also found valid in the model results.

Additionally, we also conduct robustness checks using Two-Stage Least Squares (2SLS) methods, an alternative methodology of regression following the studies of Mia [19], El-Moussawi et al. [117], Boachie [135], Abbas et al. [136], and Toh and Zhang [137]. In all the models presented in Tables 13 to 16, the First-Stage F-statistics show values greater than 10, indicating that the instruments used are relevant and not weak [19, 138]. We have also performed the weak instruments test of Stock and Yogo [139], which revealed that the First-Stage F-statistics are higher than the critical value at 5% relative bias in each model. This suggests that the instruments utilized are valid and not weak.

We re-estimate Eqs (2) to (9) using these alternative proxies (LLP and T1C) and present the results in Tables 9 to 12. We also apply the 2SLS method and show the results in Tables 13 to 16.

**Table 13. Determinants of risk and effect of capital.**

| Variable Name | Model I | Model II |
|---|---|---|
| CAR | -0.3515*** (0.0997) | -0.6485*** (0.1467) |
| CAR$^2$ | | 0.3734*** (0.0963) |
| Inflection Point | | 0.8683 |
| CFI | -1.468*** (0.4382) | -1.3433*** (0.3943) |
| DT | 0.0134** (0.0064) | 0.0103** (0.0052) |
| Size | -0.0399** (0.0176) | -0.0515*** (0.018) |
| ROA | -0.0121*** (0.0032) | -0.0058** (0.0026) |
| BSD | -0.0025* (0.0015) | -0.0016* (0.001) |
| GDP | 0.0035* (0.0021) | 0.0037** (0.0175) |
| Inflation | -0.1646** (0.0814) | -0.1142** (0.0533) |
| Constant | 1.4602** (0.7264) | 1.3836* (0.7419) |
| Year Fixed | Yes | Yes |
| Bank Fixed | Yes | Yes |
| R Squared | 0.8957 | 0.9001 |
| First-Stage *F*-Statistics | 81.7203 | 42.6502 |
| Critical Value at 5% Relative Bias | 20.74 | 20.74 |
| Observations | 450 | 450 |

Notes: The table displays the empirical results obtained from the Two-stage least squares (2SLS) with year and bank fixed-effects controlled. The dependent variable is the NPL (non-performing loans) to total loans. Standard errors are in parentheses. ***, **,* denotes the significance level at the corresponding 1%, 5%, and 10% levels. The examination of multicollinearity across all models, utilizing the VIF test, consistently revealed VIF values below the threshold of 10, indicating a negligible propensity for multicollinearity. Furthermore, the White test rejected the null hypothesis of heteroscedasticity.

The results in these tables support our main findings reported in Tables 5 through 8, with some exceptions. (See, e.g., [67, 127]).

## 5 Conclusions and policy implications

Numerous studies have delved into the connection between risk and capital, the effects of competition on the correlation between risk and capital, and the effects of ownership on the risk-capital nexus. However, the impact of competition and ownership on the simultaneous association between risk and capital has been overlooked. To address this gap, we examine how competition and ownership mediate and jointly affect the concurrent association between bank risk and capital in Bangladesh's commercial banks.

This study reveals a noteworthy association between bank risk and capital, which operates in both directions and is non-linear. The results indicate an optimum capital level that can help mitigate risk. However, our research does not guide determining this perfect capital level. The study suggests that banks with elevated risk profiles may face challenges maintaining the

**Table 14. Effect of capital, competition, and ownership on risk.**

| Variable Name | Model I | Model II | Model III | Model IV | Model V | Model VI | Model VII | Model VIII | Model IX | Model X |
|---|---|---|---|---|---|---|---|---|---|---|
| CAR | -0.3006*** (0.1045) | -0.419*** (0.1095) | -0.3007*** (0.1045) | -0.5766** (0.231) | -0.2902*** (0.0786) | -0.2482*** (0.0548) | -0.2773*** (0.0755) | -0.2793*** (0.0758) | -0.2805*** (0.0753) | -0.2793*** (0.0468) |
| CFI | -1.4719*** (0.4446) | -1.4088*** (0.4283) | -1.472*** (0.4446) | -1.4624*** (0.4222) | -1.2686*** (0.4184) | -1.4763*** (0.2281) | -1.477*** (0.4507) | -1.4822*** (0.4502) | -1.4811*** (0.4462) | -1.4802*** (0.2274) |
| DT | 0.0144** (0.0064) | -0.0138** (0.0065) | 0.0145** (0.0064) | 0.0134** (0.0067) | 0.0153*** (0.0058) | 0.0151*** (0.0054) | 0.0149** (0.0061) | 0.015** (0.0061) | 0.0149** (0.0061) | 0.0149*** (0.0054) |
| Size | -0.033* (0.0174) | -0.0368** (0.0167) | -0.033* (0.0174) | -0.0341** (0.0157) | -0.0351** (0.0157) | -0.0299*** (0.0107) | -0.0297* (0.0152) | -0.0298** (0.0151) | -0.0301** (0.015) | -0.03*** (0.0107) |
| ROA | -0.0129*** (0.0034) | -0.0126*** (0.0031) | -0.013*** (0.0034) | -0.0097*** (0.0028) | -0.0124*** (0.0036) | -0.0136*** (0.0025) | -0.0133*** (0.0036) | -0.0132*** (0.0036) | -0.0133*** (0.0036) | -0.0133*** (0.0025) |
| BSD | 0.0019 (0.0016) | -0.0016 (0.0032) | -0.0029** (0.0015) | -0.003** (0.0012) | -0.0165*** (0.0046) | -0.0044** (0.0019) | -0.0031* (0.0018) | -0.0032* (0.0018) | -0.0037* (0.002) | -0.0037** (0.0015) |
| Boone | -0.0648* (0.0364) | | | | | | | | | |
| Boone×CAR | | -0.0197* (0.011) | | | | | | | | |
| POD | | | -0.23*** (0.0652) | | | | | | | |
| POD×CAR | | | | -0.3243* (0.0185) | | | | | | |
| POD×Boone | | | | | -0.0104*** (0.0027) | | | | | |
| POD×Boone×CAR | | | | | | -0.0213** (0.0097) | | | | |
| IOD | | | | | | | -0.0093** (0.0037) | | | |
| IOD×CAR | | | | | | | | -0.1367** (0.067) | | |
| IOD×Boone | | | | | | | | | -0.0026** (0.001) | |
| IOD×Boone×CAR | | | | | | | | | | -0.0711** (0.0278) |
| GDP | 0.0045* (0.0024) | 0.0077** (0.0039) | 0.0035* (0.0018) | 0.0036* (0.002) | 0.0108** (0.005) | 0.0022 (0.0034) | 0.0035* (0.0021) | 0.0052* (0.0026) | 0.0028 (0.0031) | 0.0029 (0.0032) |
| Inflation | -0.1061* (0.0628) | -0.155* (0.0837) | -0.1822* (0.1374) | -0.1813 (0.1429) | -1.2239*** (0.3398) | -0.2903** (0.1457) | -0.1909 (0.1372) | -0.1935 (0.137) | -0.2388 (0.1457) | -0.2334** (0.1182) |
| Constant | 1.0644* (0.5631) | -0.1936 (0.2016) | 1.6809** (0.7176) | 1.4455* (0.7528) | 6.7775*** (1.7272) | 1.952*** (0.7505) | 1.4367** (0.7251) | 1.4415** (0.7248) | 1.704** (0.7728) | 1.6737*** (0.6195) |
| Year Fixed | Yes | Yes | Yes | Yes | Yes | Yes | Yes | Yes | Yes | Yes |
| Bank Fixed | Yes | Yes | Yes | Yes | Yes | Yes | Yes | Yes | Yes | Yes |
| R Squared | 0.8957 | 0.8978 | 0.8958 | 0.8987 | 0.9058 | 0.8955 | 0.8956 | 0.895 | 0.8963 | 0.8961 |
| First-Stage *F*-Statistics | 93.2117 | 51.9545 | 92.211 | 64.1189 | 103.395 | 107.258 | 105.453 | 96.1678 | 103.664 | 102.32 |
| Critical Value at 5% Relative Bias | 21.01 | 21.01 | 21.01 | 21.01 | 20.9 | 20.9 | 20.09 | 21.01 | 20.9 | 21.01 |
| Observations | 450 | 450 | 450 | 450 | 450 | 450 | 450 | 450 | 450 | 450 |

Notes: The table displays the empirical results obtained from the Two-stage least squares (2SLS) with year and bank fixed-effects controlled. The dependent variable is the NPL (non-performing loans) to total loans. Standard errors are in parentheses. ***, **, * *denotes the* significance level at the corresponding 1%, 5%, and 10% level. The examination of multicollinearity across all models, utilizing the VIF test, consistently revealed VIF values below the threshold of 10, indicating a negligible propensity for multicollinearity. Furthermore, the White test rejected the null hypothesis of heteroscedasticity.

**Table 15. Determinants of capital and effect of risk.**

| Variable Name | Model I | Model II |
|---|---|---|
| NPL | -1.2841*** (0.2412) | -1.7381* (0.9294) |
| NPL$^2$ | | -0.8977* (0.4195) |
| CFI | 0.0301** (0.0123) | 0.7137* (0.4009) |
| Lever | -0.2676** (0.1334) | -0.2691* (0.1431) |
| Size | -0.2523*** (0.0533) | -0.2411*** (0.0543) |
| ROA | 0.0135** (0.0059) | 0.0021* (0.0011) |
| BSD | -0.0015* (0.0009) | -0.0012* (0.0006) |
| GDP | -0.0035** (0.0016) | -0.0011** (0.0005) |
| Inflation | -0.1436*** (0.0247) | -0.1518*** (0.0262) |
| Constant | 4.5052*** (0.8254) | 4.4504*** (0.8551) |
| Year Fixed | Yes | Yes |
| Bank Fixed | Yes | Yes |
| R Squared | 0.7728 | 0.7969 |
| First-Stage *F*-Statistics | 64.6189 | 78.2988 |
| Critical Value at 5% Relative Bias | 20.74 | 20.9 |
| Observations | 450 | 450 |

Notes: The table displays the empirical results obtained from the Two-stage least squares (2SLS) with year and bank fixed-effects controlled. The dependent variable is the CAR (Capital Adequacy Ratio). Standard errors are in parentheses. ***, **, * *denotes the* significance level at the corresponding 1%, 5%, and 10% level. The examination of multicollinearity across all models, utilizing the VIF test, consistently revealed VIF values below the threshold of 10, indicating a negligible propensity for multicollinearity. Furthermore, the White test rejected the null hypothesis of heteroscedasticity.

required capital adequacy. These findings align with the moral hazard and regulatory hypotheses.

This study has demonstrated strong evidence of the effect of competition on the simultaneous relationship between risk and capital. This study reveals that banks tend to hold larger capital levels in highly contested markets. Moreover, heightened competition has been found to lessen the risk appetite of Bangladeshi commercial banks. These results support the competition stability hypothesis (e.g., [33]). Additionally, our findings indicate that maintaining elevated regulatory capital levels can effectively curtail credit risks for banks operating in competitive markets.

The study provides compelling evidence of the effect of ownership on the relationship between risk and capital. Interestingly, the research indicates that private and Islamic commercial banks are more risk-averse than state-owned and conventional commercial banks. This aligns with the "political view of state ownership," as state-owned banks often have political ties that influence their decision-making. Furthermore, private and conventional banks are likely to hold more capital and take fewer risks in competitive markets. State-owned banks and

**Table 16. Effect of risk, competition, and ownership on capital.**

| Variable Name | Model I | Model II | Model III | Model IV | Model V | Model VI | Model VII | Model VIII | Model IX | Model X |
|---|---|---|---|---|---|---|---|---|---|---|
| NPL | -1.2605*** (0.3005) | -1.2086*** (0.2417) | -0.5304*** (0.1478) | -1.2977*** (0.352) | -0.7909*** (0.14) | -0.9132*** (0.0847) | -0.7056*** (0.0749) | -0.7052*** (0.0741) | -1.2186*** (0.2845) | -1.1441*** (0.2742) |
| CFI | 1.2764*** (0.4026) | 1.0197*** (0.3025) | 0.6908** (0.2842) | 1.0639*** (0.3443) | 1.0327** (0.4196) | 1.593*** (0.3581) | 1.0749*** (0.319) | 1.1104*** (0.3164) | 2.1825*** (0.6957) | 2.0245*** (0.6799) |
| Lever | -0.1316** (0.0609) | -0.1106** (0.0493) | -0.1399** (0.0703) | -0.155** (0.0689) | -0.1912* (0.1098) | -0.1289*** (0.0401) | -0.138*** (0.0363) | -0.1373*** (0.036) | -0.245** (0.1129) | -0.1331 (0.1143) |
| Size | -0.1477*** (0.0334) | -0.1623*** (0.0342) | -0.1339*** (0.03) | -0.1474*** (0.0336) | -0.1433*** (0.0298) | -0.1414*** (0.0138) | -0.137*** (0.0126) | -0.1395*** (0.0125) | -0.1464*** (0.033) | -0.1452*** (0.0324) |
| ROA | 0.0058 (0.0065) | 0.0022 (0.006) | 0.0084* (0.0048) | 0.0023* (0.013) | 0.0091** (0.0045) | 0.0013 (0.0033) | 0.005* (0.003) | 0.0056* (0.0029) | -0.0051 (0.0063) | -0.0036 (0.0064) |
| BSD | -0.0031* (0.0018) | -0.0126*** (0.0022) | -0.0043*** (0.0014) | -0.0011** (0.0005) | -0.0091* (0.0046) | -0.0043** (0.0019) | 0.0034** (0.0016) | -0.0035** (0.0016) | -0.0002 (0.0027) | -0.0011 (0.0026) |
| Boone | 0.0025* (0.0015) | | | | | | | | | |
| Boone×NPL | | -0.061*** (0.0139) | | | | | | | | |
| POD | | | -0.0153* (0.0081) | | | | | | | |
| POD×NPL | | | | 0.5699* (0.3311) | | | | | | |
| POD×Boone | | | | | 0.0096*** (0.0029) | | | | | |
| POD×Boone×NPL | | | | | | 0.0218** (0.0097) | | | | |
| IOD | | | | | | | -0.0608* (0.0364) | | | |
| IOD×NPL | | | | | | | | -0.7415** (0.2925) | | |
| IOD×Boone | | | | | | | | | -0.0031* (0.0016) | |
| IOD×Boone×NPL | | | | | | | | | | -0.1468** (0.0652) |
| GDP | -0.0034** (0.0017) | -0.0173*** (0.0052) | -0.0029** (0.0013) | -0.0058** (0.0024) | -0.0087* (0.0051) | -0.0073* (0.0039) | 0.0062* (0.0033) | -0.0041 (0.0034) | -0.0084* (0.0048) | -0.0065* (0.0038) |
| Inflation | -0.2589** (0.1138) | -0.8930*** (0.1867) | -0.2062* (0.106) | 0.0582 (0.1783) | -0.7933** (0.3433) | -0.267* (0.1493) | -0.1659 (0.1236) | 0.1788 (0.1227) | 0.0083 (0.197) | 0.0537 (0.1853) |
| Constant | 3.5461*** (0.7874) | 2.1228** (0.8618) | 1.018* (0.5723) | 1.9254* (1.0016) | 6.2295*** (1.8214) | 0.7835 (0.7806) | 1.2391* (0.652) | 1.1675* (0.6464) | 2.26** (1.1215) | 1.928* (1.0474) |
| Year Fixed | Yes | Yes | Yes | Yes | Yes | Yes | Yes | Yes | Yes | Yes |
| Bank Fixed | Yes | Yes | Yes | Yes | Yes | Yes | Yes | Yes | Yes | Yes |
| R Squared | 0.6891 | 0.781 | 0.8494 | 0.6999 | 0.8305 | 0.7938 | 0.8299 | 0.8325 | 0.8309 | 0.7287 |
| First-Stage F-Statistics | 87.0379 | 75.3714 | 79.643 | 72.1899 | 63.661 | 86.2182 | 84.5893 | 84.7814 | 94.7333 | 87.5132 |
| Critical Value at 5% Relative Bias | 20.9 | 20.9 | 21.01 | 20.9 | 21.01 | 21.01 | 21.18 | 21.18 | 21.01 | 21.1 |
| Observations | 450 | 450 | 450 | 450 | 450 | 450 | 450 | 450 | 450 | 450 |

Notes: The table displays the empirical results obtained from the Two-stage least squares (2SLS) with year and bank fixed-effects controlled. The dependent variable is the CAR (Capital Adequacy Ratio). Standard errors are in parentheses. ***, **,* denotes the significance level at the corresponding 1%, 5%, and 10% levels. The examination of multicollinearity across all models, utilizing the VIF test, consistently revealed VIF values below the threshold of 10, indicating a negligible propensity for multicollinearity. Furthermore, the White test rejected the null hypothesis of heteroscedasticity.

Islamic banks experience a decrease in their capital base due to increased risks. The study finds that private and Islamic banks take fewer risks in highly competitive markets, while state-owned and conventional banks take greater risks. Again, this study postulates that private and

Islamic banks with strong capital positions in a competitive market can better manage risks than state-owned and conventional banks. However, it is also noted that maintaining higher capital ratios in competitive environments proves to be a challenge for state-owned and Islamic banks as compared to private-owned and conventional commercial banks.

The findings of this research carry substantial weight for policymakers in Bangladesh, particularly in developing countries like Bangladesh. The study has revealed a significant association between competition, capital, and credit risk. As a result, policymakers must consider promoting a balanced-competitive banking market, as this can lead to higher regulatory capital levels and effectively reduce credit risks for banks operating in such markets. To manage their capital and monitor non-performing loans effectively, Bangladeshi banks must improve their monitoring policies. Policymakers must also exercise caution with government banks and ensure they do not invest in risky projects. Moreover, policymakers may consider encouraging state-owned and conventional banks, promoting competition in the banking sector, and highlighting the need for solid financial foundations. Furthermore, policymakers might examine the challenges that state-owned and Islamic banks face in competitive and risky environments to ensure the stability and resilience of the banking sector. Researchers could utilize other econometric models or methodologies, such as machine learning techniques, to analyze the relationship between capital regulation and risk-taking. Comparative studies involving other developing countries can provide broader insights.

To sum up, this paper investigates the impact of competition and ownership on the risk-capital nexus within a single country without considering the cross-country effect. Moreover, this paper cannot incorporate all banks operating in Bangladesh, as FCBs and SDBs were excluded due to non-availability and inconsistent reporting, and banks with less than five consecutive annual reports were also removed. Additionally, it is important to recognize that the findings of this study are contextually specific to the Bangladeshi banking sector and may not be directly applicable to other countries or regions with different market dynamics, regulatory environments, and economic conditions. Nonetheless, we intend to delve into this aspect in subsequent research.

## Supporting information

**S1 Appendix.**
(DOCX)

**S1 Dataset.**
(XLSX)

## Acknowledgments

This research received no grants from public, commercial, or not-for-profit sectors.

## Author Contributions

**Conceptualization:** Changjun Zheng, Md Mohiuddin Chowdhury, Anupam Das Gupta.

**Data curation:** Md Mohiuddin Chowdhury, Anupam Das Gupta, Md Nazmul Islam.

**Formal analysis:** Md Mohiuddin Chowdhury.

**Methodology:** Md Mohiuddin Chowdhury, Anupam Das Gupta, Md Nazmul Islam.

**Software:** Md Mohiuddin Chowdhury.

**Supervision:** Changjun Zheng.

**Validation:** Changjun Zheng.

**Writing – original draft:** Md Mohiuddin Chowdhury.

**Writing – review & editing:** Md Mohiuddin Chowdhury, Anupam Das Gupta, Md Nazmul Islam.

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
