## [Decision Letter · Decision Letter 0]

3 Apr 2024

PONE-D-24-05802Effects of competition and ownership on the simultaneous relationship between bank risk and capital:  evidence from an emerging economyPLOS ONE

Dear Dr. Chowdhury,

Thank you for submitting your manuscript to PLOS ONE. After careful consideration, we feel that it has merit but does not fully meet PLOS ONE’s publication criteria as it currently stands. Therefore, we invite you to submit a revised version of the manuscript that addresses the points raised during the review process.

**Based on the comments below. Kindly revise the manuscript. Provide line by line response to each of the review with a track changed manuscript. Note that without addressing proper changes and response to reviews, the manuscript will not be proceeded further.**

We look forward to receiving your revised manuscript.

Kind regards,

Dr. Muhammad Usman Tariq

PFHEA, CFCIPD, CMBE, SFSEDA

Academic Editor

PLOS ONE

Journal Requirements:

2. In the online submission form, you indicated that Data will be made available on request.

3. We note that you have referenced Berger, A.N., Ö. Öztekin, and R.A. Roman which has currently not yet been accepted for publication. Please remove this from your References and amend this to state in the body of your manuscript: (Berger, A.N., Ö. Öztekin, and R.A. Roman [Unpublished]) as detailed online in our guide for authors

Reviewers' comments:

Reviewer's Responses to Questions

**Comments to the Author**

1. Is the manuscript technically sound, and do the data support the conclusions?

Reviewer #1: Yes

Reviewer #2: Yes

Reviewer #3: Partly

Reviewer #4: Partly

Reviewer #5: Yes

Reviewer #6: Yes

Reviewer #7: Yes

2. Has the statistical analysis been performed appropriately and rigorously? 

Reviewer #1: Yes

Reviewer #2: Yes

Reviewer #3: Yes

Reviewer #4: Yes

Reviewer #5: Yes

Reviewer #6: Yes

Reviewer #7: Yes

3. Have the authors made all data underlying the findings in their manuscript fully available?

Reviewer #1: Yes

Reviewer #2: Yes

Reviewer #3: Yes

Reviewer #4: Yes

Reviewer #5: Yes

Reviewer #6: No

Reviewer #7: Yes

4. Is the manuscript presented in an intelligible fashion and written in standard English?

Reviewer #1: No

Reviewer #2: Yes

Reviewer #3: Yes

Reviewer #4: Yes

Reviewer #5: Yes

Reviewer #6: Yes

Reviewer #7: No

5. Review Comments to the Author

**Reviewer #1**: Recommendations for Manuscript ID PONE-D-24-05802 Title: „Effects of competition and ownership on the simultaneous relationship between bank risk and capital: evidence from an emerging economy” for the Plos One Journal.

General Comments

From my point of view, it is a very interesting topic and simultaneously it seems that to the best of my knowledge is the first empirical study employs the two-step-system generalized method of moments to examine the effect of competition and ownership on the simultaneous association between risk and capital. The findings depict that competition and ownership significantly impact the relationship between risk and capital, which aligns with the moral hazard hypothesis, competition fragility hypothesis, and political view of state ownership.

The paper contains the following sections: Introduction, Literature Review, Data, Variables, and Methodology of the study, Empirical Research Framework, Empirical Findings and Discussions, and Conclusion.

However, I find some recommendations:

1. I suggest to the authors that the last section Conclusions and Policy Implications.

2. The abstract must contain the main purpose of the paper, the research method used in the research and the main contributions.

3. It would be very useful to add the "Introduction" section and the purpose, objectives and hypothesis of the research. I consider that a weak point of the paper is that the authors did not show the novelty of the paper compared to other works. That is why, I consider that the introduction should specify the novelty of the paper compared to other papers published in this area.

4. The research is well based on science and the results are in agreement with the theoretical part. From my point of view, the paper is original and the topic addressed brings added value to the specialized literature regarding financial markets. The paper is well written and easy to read.

5. The research is well based on science and the results are in agreement with the theoretical part. The model applied to the analyzed data is correctly used in the analysis undertaken, it is a strength point of this paper.

6. The authors must specify the software used (STATA, Eviews, SPSS, etc.). Then the authors must do the Hausman test to detect the cross section with fixed or random effects. It must show the results of the Likelihood test, the heteroskedasticity test, and the endogeneity test.

7. The authors do not present the working hypotheses and do not use the standardized mathematical formulas for the econometric equations.

8. It is very important for the authors to analyze the descriptive analysis (with Kurtosis test, Jarque Berra test and interpretation, Skewness and Kurtosis interpretation). In the same time the correlation analysis and the VIF test are very important in this research.

9. Obviously, the application of a model must take into account the tests of heteroskedasticity, endogeneity etc. These aspects do not emerge from the method and results. In the same time, in the Robustness Checks section the authors have to apply an econometric method like regression or panel with fixed effect estimation or the random effect estimation (see for instance, Baltagi (2008), Hsiao (2014) and Andre B et al. (2015)). Besides, the corresponding tests to determine which is the best method of estimation is needed (see the Hausman test, the Breusch and Pagan (1980)´s Lagrange multiplier, the F test for fixed effects to test whether all unobservable individual effects are zero).

10. The authors talk about the relationship between these variables, however they do not support the empirical evidence providing panel cointegration tests that are crucial (see for instance Kao (1999) panel data cointegration test, the Pedroni (1999, 2004) panel data cointegration test or the Westerlund (2005) panel data cointegration test, among others.

11. The authors must show which are the instruments used by the 2SLS method. At the same time, the authors must explain when the null hypotheses of heteroskedasticity and endogeneity are accepted and what are the tests used in this regard.

12. I recommend the authors to refer to other recent works indexed in Web of Science, because only some cited works is not enough for a scientific paper. In my opinion, the authors must cite other papers regarding this subject and other subjects such as: tax compliance, banking, economic growth etc. That is why, I suggest that the authors cite papers published in Web of Science Journals, such as:

1. Batrancea, L., Rathnaswamy, M.K., & Batrancea, I. (2021). A Panel Data Analysis on Determinants of Economic Growth in Seven Non-BCBS Countries. Journal of the Knowledge Economy, 13, 1651–1665.

2. Batrancea, L., Rathnaswamy, M.M., Rus, M.S., & Tulai, H. (2022). Determinants of Economic Growth for the Last Half of Century: A Panel Data Analysis on 50 Countries. Journal of the Knowledge Economy.

3. Balcı, M.A., Batrancea, L.M., Akgüller, Ö., & Nichita, A. (2022). Coarse Graining on Financial Correlation Networks. Mathematics, 10(12), 2118.

4. Balcı, M.A., Batrancea, L.M., Akgüller, Ö., Gaban, L., Rus, M.-I., & Tulai, H. (2022). Fractality of Borsa Istanbul during the COVID-19 Pandemic. Mathematics, 10(14), 2503.

5. Batrancea, I., Batrancea, L., Rathnaswamy, M.M., Tulai, H., Fatacean, G., & Rus, M.-I. (2020). Greening the Financial System in USA, Canada and Brazil: A Panel Data Analysis. Mathematics, 8(12), 2217.

6. Batrancea, L. (2021). An Econometric Approach on Performance, Assets, and Liabilities in a Sample of Banks from Europe, Israel, United States of America, and Canada. Mathematics, 9(24), 3178.

7. Batrancea, L.M. Determinants of Economic Growth across the European Union: A Panel Data Analysis on Small and Medium Enterprises. Sustainability 2022, 14, 4797. https://doi.org/10.3390/su14084797.

8. Batrancea, L.M.; Balcı, M.A.; Chermezan, L.; Akgüller, Ö.; Masca, E.S.; Gaban, L. Sources of SMEs Financing and Their Impact on Economic Growth across the European Union: Insights from a Panel Data Study Spanning Sixteen Years. Sustainability 2022, 14, 15318. https://doi.org/10.3390/su142215318

9. Batrancea, L.M.; Balcı, M.A.; Akgüller, Ö.; Gaban, L. What Drives Economic Growth across European Countries? A Multimodal Approach. Mathematics 2022, 10, 3660. https://doi.org/10.3390/math10193660.

10. Batrancea, L.; Pop, M.C.; Rathnaswamy, M.M.; Batrancea, I.; Rus, M.-I. An Empirical Investigation on the Transition Process toward a Green Economy. Sustainability 2021, 13, 13151. https://doi.org/10.3390/su132313151

11. Batrancea, L., Batrancea, I., & Moscviciov, A. (2009). The Roots of the World Financial Crisis. Annals of the University of Oradea. Economic Sciences, 3, 57–62.

12. Moscviciov, A., Batrancea, I., Batrancea, M., Batrancea, L. (2010) Financial Ratio Analysis Used In The It Enterprises, Annals of Faculty of Economics, 1(2), 600-603.

In conclusion, the article should be improve. It should also be enhanced with a review of the literature adequate to the subject and a broader interpretation and commentary of the research results.

**Reviewer #2:** Overall, the paper is an interesting read and has considerable scope. By considering the feedback provided and using the recommended references, the authors can enhance the study's impact and attract more interest. In short, I see this paper has potential to be considered for publication after incorporating the following suggestions.

Main Concern :

• I suggest the author add separate paragraphs of your study contribution citing papers to support their claim. I suggest that the contribution should be rewritten. In the current versions, there are no strong contributions. You should add here some details on the contribution of this paper as compared to previous literature and how it is different. In study contribution author(s) should cite past papers to support their claim.

Example: https://doi.org/10.1016/j.ribaf.2024.102305

This article contributes to the related literature in the following ways. First, according to the best of our knowledge, this study provides new cross-country evidence by exploring the non-linear patterns of ESG performance on sustainable growth rate of listed firms in the United States and China. This study is different from previous studies, such as (Chai et al., 2023, Fuente et al., 2022, Teng et al., 2021) adds a novel perspective to the existing body of research. Second, we analyze the non-linear effect of individual components on environmental, social, and governance performance and their distinct impact on sustainable growth. Third, we utilize the widely recognized measures of firms' sustainable growth rate models namely Robert Higgins widely employed in the domains of strategic management and corporate planning. Methodologically, we apply High Dimension Fixed Effects, feasible generalized least square, and dynamic panel two-stage generalized method of moments models to analyze the data. In addition, heterogeneous subgroups, and alternative model SG were developed by Van Horn. Furthermore, the exploration of the threshold effect of overall ESG performance, coupled with the analysis of individual environmental, social, and governance performance, introduces a fresh dimension to academic discussions.

Literature review

• The following papers will help you in building theoretical farmwork, motivation, and in gap justification. Authors must improve the quality of references and add recent articles, such as.

https://doi.org/10.1016/j.ribaf.2024.102305

https://doi.org/10.1016/j.heliyon.2024.e26757

https://doi.org/10.1016/j.bir.2024.01.002

https://doi.org/10.24136/oc.2023.036

• This paper lacks a good knowledge of the key theories. Also, it is important to show the implications of these theories for your study.

• I suggest authors to conclude literature section follow Example:

“Drawing from the study literature analysis, we believe that the interplay between ESG performance, traditional firm characteristics, industry dynamics, macroeconomic factors, and financial determinants in shaping corporate capital structure finds theoretical support from various established theories in corporate finance. In analyzing corporate capital structure in BRICS economies, several theories provide theoretical underpinnings and support. The trade-off theory suggests that firms balance the benefits of debt financing, such as tax shields, against the costs, such as bankruptcy risk, to determine an optimal capital structure (Miller & Modigliani, 1961). Agency theory highlights the conflicts of interest between stakeholders and managers, leading to the monitoring and disciplining role of debt in aligning interests (Meckling & Jensen, 1976). The pecking order theory posits that firms prefer internal financing over external debt due to information asymmetry, followed by debt over equity issuance(Myers, 1984a). Additionally, stakeholder theory emphasizes the importance of considering the interests of all stakeholders, including environmental and social concerns, in financial decision-making (Freeman, 1984). These theories provide a framework for understanding how ESG factors, firm characteristics, industry dynamics, macroeconomic conditions, and financial factors collectively influence corporate capital structure in BRICS economies. Drawing on well-established theories in corporate finance, such as stakeholder theory, TOT and POT , the study conceptual framework has been visualized in Figure 1”.

Data and Methodology

• In the current section data snad Sample constraction detial is complely missing ? I suggest authors to add one full set infirmation about data sourcses and period of smaple data and sample criteria. I am suggesting following papers to follow how to write this section

https://doi.org/10.1016/j.ribaf.2024.102305

https://doi.org/10.1016/j.heliyon.2024.e26757

https://doi.org/10.1016/j.bir.2024.01.002

https://doi.org/10.24136/oc.2023.036

• I have a concern regarding the explanation of the concept of "Inflection Point" as mentioned in line 403. While the formula provided sheds some light on the concept, I believe that a more thorough explanation is necessary to ensure clarity for the readers. Simply presenting the formula may not fully serve the purpose of elucidating the concept. I would suggest expanding upon the explanation of the "Inflection Point," providing more context and possibly offering examples to illustrate its significance within the context of your study.

Empirical results

• In my opinion, the authors should provide a more detailed discussion and provide economic justification /connectivity of findings. Drawing from past empirical studies and theoretical frameworks would not only enrich their analysis but also lend credibility to their arguments.

• Conclusion

• Policy Implication should be improved. The authors should add more theoretical contributions and implications for managerial practice. These discussions could be more specific and thoughtful (take some practical examples).

• Add study limitations based on your study and at least one open research questions for future researchers.

Minor revision

• On a final note, I would recommend language editing. The references should be presented according to journal guidelines. In most of the reference volumes the page number and issue are missing.

• Ensure that each table has explanatory notes to make it self-explanatory. This will improve the clarity and accessibility of the presented data for readers.

Best of luck

**Reviewer #3**: This paper addresses the important topic of the effects of competition and ownership on the relation between bank risk and capital in Bangladesh and has the merit of addressing and emerging country often neglected in literature, therefore filling an important research gap. However, some improvements might be needed to meet the journal standards. Thus, my comments to the authors are as follows:

1. To provide a comprehensive description of the data the skewness, kurtosis and Jarque-Bera test should also be addressed.

2. Authors should perform autocorrelation and heteroskedasticity tests, which are missing in the paper and are important to examine the adequacy of the model estimates.

3. As authors provide the estimates of ten models it would be interesting to understand how to discriminate between them. What is the best model to describe the data.

4. finally, the residual analysis is not provided. Therefore, I suggest that it should be included in the paper.

**Reviewer #4**: Comments, Suggestions and Questions

1.Topic

The topic is Very interesting but it is better to state study area like

“Effects of competition and ownership on the simultaneous relationship between bank risk and capital: evidence from an emerging economy of Bangladesh”.

2.Abstract

1.You did not mention the objective of the study, crucial background, So, clearly describe it.

2.You did not mention the research methodology that you used, so clearly state your research methodology that you used.

The reset section of the abstract is good.

3.Introduction

Your introduction sections are strong and you are clearly articulate the research problem and its relevance.

However, when you state your research gap you have stated only title gap and variable gap. So clearly mention the other relevant research gaps that you are interested to do this research topic (clear gap analysis and methods used by other studies)

4.Theoretical and Empirical literature review

1.You are mentioning the empirical review of the previous researcher main finding and develop hypothesis, but you did not Clearly incorporate the variables that are used, the methodology they used in the empirical review section?

2.You have put dependent and independent variables under your variable description part, so it is better to put by using the conceptual framework to show its relationship.

3.You have described to use different theories under paragraph 5 of line 89 and 90 in your introduction part but, you are not discussing these theories in detail so, you should to discuss by relating to your research topic.

4.Your empirical literature review is not link with study area that is Bangladesh needed to be provided related literatures.

5.Methodology

1.You did not mention the research design and sampling technique that you used.

2.You are not mentioning the model that you are used and the reason?

3.The rationale for the chosen time period 2010 to 2021 for data collection needs further explanation. Discuss how this period captures relevant economic cycles, regulatory changes, or other factors affecting computation and ownership of banks.

6.Empirical finding and Discussion

The discussion part is well discussed. You are discussed each independent variable relationship with the independent variable, so it is good.

7.Conclusions

Your conclusion is somewhat good but, you should briefly state your thoughts, your main points from your analysis and interpretation.

8.Recommendation

You only give recommendation for policy makers, you forget to mention direction for future researcher, banks and others.

Reference…It is very good. I have not seen a problem it is well citied and you have use different references.

**Reviewer #5:** Reviewer 1 Comments:

The paper mentioned above is a good effort by the researcher(s); however, the following points need consideration.

1) The abstract is rather technical, please try to explain clearly. The findings in the abstract need to be written in a more informative way.

2) Motivation and contribution need to be strengthened. The contribution of this study needs more logical detail. The contribution of this paper is not well explained in the current version of the paper.

3) The authors should reorganize the structure of the introduction section to thoroughly express the aspects of this study including the background, current progress, motivation, the research question, the objective, the contribution, etc.

4) The introduction of the study is too short, even references are not enough to justify the topic. Please follow and cite some recent research papers such as;

I. https://doi.org/10.1016/j.jup.2024.101719

II. https://doi.org/10.1007/s10614-022-10353-4

III. https://doi.org/10.1016/j.egyr.2022. 11.054 2352-4847

IV. https://doi.org/10.1016/j.renene.2023.03.090

V. https://doi.org/10.1007/s10668-023-03843-4

5) The literature review is so lengthy unnecessary studies could be removed. Moreover, the manuscript could be substantially improved by relying and citing more on recent literature 2022-2023 such as the followings:

i. https://doi.org/10.1007/s10644-021-09321-z

ii. https://doi.org/10.1016/j.esr.2022.1008

iii. https://doi.org/10.1007/s10668-023-03413-8

6) All equations need appropriate references in the text. Equation 2. is not appropriately referenced in the text.

7) Include the mean value of VIF in the last row in the Table. 3.

8) Divide your article into clearly defined and numbered sections. Subsections should be numbered 1.1 (then 1.1.1, 1.1.2, ...), 1.2, etc.

9) In the conclusion part, the authors should list all the limitations and discuss possible directions for further research.

10) Please check the manuscript again for errors.

**Reviewer #6**: • The research results in the Abstract should present the specific relationship between bank capital and risk.

• Hypothesis 1 and 2 should clearly state the bidirectional relationship between bank risk and capital as considered in this article.

• The chapter on Theoretical and Conceptual Background can be appropriately simplified as part of the Introduction and Literature review.

• Date: The data should show how many state-owned, private, traditional, and Islamic banks each have as samples.

• The reasons for the inconsistency between the empirical results of this article and existing literature research should be explained, such as line 599.

• The regression results in each table should be reported as standard errors rather than P-values.

**Reviewer #7**: - The names of the variables (i.e., “endo”) in some equations seem irrelevant and difficult to follow. The authors should change the notation of the variables.

- Equation 11 (inflection point) is not necessary.

- The description of variables in Table 1 should be described by words rather than by formula.

- The inverse of b is 1/b, not -b. The author should correct the use of word.

- The authors should check for typos and errors such as “by utilizing different specifications of the dependent and endogenous independent variables and the dependent and endogenous independent variables.”

- The authors should describe the instrumental variables in the robustness check in more details.

- The research hypotheses should be better developed and revised.

- The inclusion of inflection points in the estimation results is not necessary and unsuitable.

- The references are not sorted and have considerable errors.

6. PLOS authors have the option to publish the peer review history of their article (what does this mean?). If published, this will include your full peer review and any attached files.

Reviewer #1: No

Reviewer #2: No

Reviewer #3: No

Reviewer #4: No

Reviewer #5: **Yes: **Muhammad Akhtar

Reviewer #6: No

Reviewer #7: No

---

## [Author Response · Author response to Decision Letter 0]

24 Aug 2024

Original Manuscript ID: PONE-D-24-05802

Original Article Title: "Effects of competition and ownership on the simultaneous relationship between bank risk and capital: evidence from an emerging economy" 

General comments from the Honorable Editor

Dear Dr. Chowdhury,

Thank you for submitting your manuscript to PLOS ONE. After careful consideration, we feel that it has merit but does not fully meet PLOS ONE’s publication criteria as it currently stands. Therefore, we invite you to submit a revised version of the manuscript that addresses the points raised during the review process.

Response to Honorable Editor:

Dear Sir, Thank you very much for allowing us to address your valuable comments and those of the honorable reviewers on our research paper. We have addressed the comments and attached the revised paper for your kind consideration.

Thank you very much for your attention and consideration.

Kind Regards,

The Authors

Response to Editor

General Comments: 

Thank you for submitting your manuscript to PLOS ONE. After careful consideration, we feel that it has merit but does not fully meet PLOS ONE’s publication criteria as it currently stands. Therefore, we invite you to submit a revised version of the manuscript that addresses the points raised during the review process.

Authors' Response: The authors thank the honorable reviewer for the positive and valuable comments. All the suggestions and comments are thoughtfully considered and addressed in the revision.

Comment 1: 

Authors’ response: Honorable Editor, thank you for your guidance. We have carefully reviewed and ensured that our manuscript adheres to PLOS ONE’s style requirements, including proper file naming and formatting as outlined in the provided templates.

Comment 2: In the online submission form, you indicated that Data will be made available on request.

Authors’ response: Dear Editor, thank you for the clarification regarding data availability. In compliance with PLOS ONE’s requirements, we have made all data available to other researcher. 

Comment 3: We note that you have referenced Berger, A.N., Ö. Öztekin, and R.A. Roman which has currently not yet been accepted for publication. Please remove this from your References and amend this to state in the body of your manuscript: (Berger, A.N., Ö. Öztekin, and R.A. Roman [Unpublished]) as detailed online in our guide for authors

Authors’ response: Honorable Editor, thank you very much. We have updated the reference.

Comment 4: f Please include captions for your Supporting Information files at the end of your manuscript, and update any in-text citations to match accordingly. Please see our Supporting Information guidelines for more information: http://journals.plos.org/plosone/s/supporting-information. 

Authors’ response: Dear Editor, thank you for your guidance. We have included captions for all Supporting Information files at the end of the manuscript and updated the in-text citations to ensure they match accordingly. We have also reviewed the Supporting Information guidelines to ensure full compliance.

Response to Reviewers:

Reviewer 1: 

General Comments: 

Dear Editor,

From my point of view, it is a very interesting topic and simultaneously it seems that to the best of my knowledge is the first empirical study employs the two-step-system generalized method of moments to examine the effect of competition and ownership on the simultaneous association between risk and capital. The findings depict that competition and ownership significantly impact the relationship between risk and capital, which aligns with the moral hazard hypothesis, competition fragility hypothesis, and political view of state ownership.

The paper contains the following sections: Introduction, Literature Review, Data, Variables, and Methodology of the study, Empirical Research Framework, Empirical Findings and Discussions, and Conclusion. However, I find some recommendations.

Authors' Response: The authors thank the honorable reviewer for the positive and valuable comments. All the suggestions and comments are thoughtfully considered and addressed in the revision.

Comment 1: I suggest to the authors that the last section Conclusions and Policy Implications.

Authors’ response: Respected reviewer, thank you for your insightful observation. Following your observation, we renamed the conclusion “Conclusions and Policy Implications”.

Comment 2: The abstract must contain the main purpose of the paper, the research method used in the research and the main contributions.

Authors’ response: Dear Reviewer, Thank you for your valuable feedback on our manuscript. We have carefully considered your comments and made the necessary revisions to the abstract to better align with your suggestions. Specifically, we have ensured that the abstract clearly outlines the paper's main purpose, the research methods employed, and the primary contributions of our study. Please see the abstract. These changes effectively address your concerns and enhance the clarity and comprehensiveness of our abstract. Thank you again for your insightful feedback, which has significantly improved the quality of our manuscript.

Comment 3: It would be very useful to add the "Introduction" section and the purpose, objectives and hypothesis of the research. I consider that a weak point of the paper is that the authors did not show the novelty of the paper compared to other works. That is why, I consider that the introduction should specify the novelty of the paper compared to other papers published in this area.

Authors’ response: Dear revered reviewer, thank you very much. Following your comment, we add the purpose and objectives of the research in the introduction section. Please see, lines 78-80. We specify the hypotheses of the research in the literature review part. Moreover, we specify the novelty of the paper in the introduction part of the paper. Please see, lines 82-89.

Comment 4: The research is well based on science and the results are in agreement with the theoretical part. From my point of view, the paper is original and the topic addressed brings added value to the specialized literature regarding financial markets. The paper is well written and easy to read.

Authors’ response: Dear valued reviewer, thank you very much for your insightful comments and positive feedback on our research paper. We sincerely appreciate your recognition of the alignment between our research findings and the theoretical underpinnings and your acknowledgment of the paper's originality and contribution to the specialized literature on financial markets. It is encouraging to hear that you found the paper well-written and easy to read, as we aimed to ensure clarity and accessibility for our readers. We are grateful for your thoughtful assessment.

Comment 5: The research is well based on science and the results are in agreement with the theoretical part. The model applied to the analyzed data is correctly used in the analysis undertaken, it is a strength point of this paper.

Authors’ response: Thank you for your positive assessment of our research methodology and the alignment of our results with theoretical expectations. We are pleased to hear that you found the application of our analytical model appropriate and consider it a strength of the paper. Your feedback is valuable to us.

Comment 6: The authors must specify the software used (STATA, Eviews, SPSS, etc.). Then the authors must do the Hausman test to detect the cross section with fixed or random effects. It must show the results of the Likelihood test, the heteroskedasticity test, and the endogeneity test.

Authors’ response: Dear reviewer, thank you for your nice observations and comments. Following your observation, we specify which software we used for data analysis in the study. Please see the Empirical research framework section (lines 416-417). We also show the heteroscedasticity test, and the endogeneity test. Please see Appendix C.

Comment 7: The authors do not present the working hypotheses and do not use the standardized mathematical formulas for the econometric equations.

Authors’ response: Dear reviewer, thank you very much. As per your valuable suggestions, we have presented the working hypothesis (please see hypotheses 1 to 5) using the standardized mathematical formulas for the econometric equations Please see equations 2 to 11.

Comment 8: It is very important for the authors to analyze the descriptive analysis (with Kurtosis test, Jarque Berra test and interpretation, Skewness and Kurtosis interpretation). In the same time the correlation analysis and the VIF test are very important in this research.

Authors’ response: Dear Reviewer, Thank you very much for your insightful suggestions. Following your comment, we analyze the descriptive statistics with skewness, kurtosis, and the Jarque Berra Test with interpretation. Please see Table 2 and Line numbers 470 – 474.

Comment 9: Obviously, the application of a model must take into account the tests of heteroskedasticity, endogeneity etc. These aspects do not emerge from the method and results. In the same time, in the Robustness Checks section the authors have to apply an econometric method like regression or panel with fixed effect estimation or the random effect estimation (see for instance, Baltagi (2008), Hsiao (2014) and Andre B et al. (2015)). Besides, the corresponding tests to determine which is the best method of estimation is needed (see the Hausman test, the Breusch and Pagan (1980)´s Lagrange multiplier, the F test for fixed effects to test whether all unobservable individual effects are zero).

Authors’ response: We consider the multicollinearity tests (VIF) and white test for homoscedasticity for every model. 

Our diagnosis results suggested fixed effects in the regression model (please see Table C4 in Appendix C). we opt for model regression with fixed effects in both the baseline model and robust check. We also run the heteroscedasticity test, endogeneity test, and the Breusch and Pagan Lagrange multiplier test (please see Appendix C).

Comment 10: The authors talk about the relationship between these variables, however they do not support the empirical evidence providing panel cointegration tests that are crucial (see for instance Kao (1999) panel data cointegration test, the Pedroni (1999, 2004) panel data cointegration test or the Westerlund (2005) panel data cointegration test, among others.

Authors’ response: Dear Reviewer, Thank you very much. Following your valuable comments, we incorporated panel cointegration tests. Please see Lines 494 – 500, and also Tables B1 and B2 in Appendix B.

Comment 11: The authors must show which are the instruments used by the 2SLS method. At the same time, the authors must explain when the null hypotheses of heteroskedasticity and endogeneity are accepted and what are the tests used in this regard.

Authors’ response: Dear Reviewer, following the suggestion we mentioned about the instrument variables in robust check. The diagnosis tests are also incorporated with the explanation of relevant hypotheses in Appendix C.

Comment 12: I recommend the authors to refer to other recent works indexed in Web of Science, because only some cited works is not enough for a scientific paper. In my opinion, the authors must cite other papers regarding this subject and other subjects such as: tax compliance, banking, economic growth etc. That is why, I suggest that the authors cite papers published in Web of Science Journals, such as:

1. Batrancea, L., Rathnaswamy, M.K., & Batrancea, I. (2021). A Panel Data Analysis on Determinants of Economic Growth in Seven Non-BCBS Countries. Journal of the Knowledge Economy, 13, 1651–1665.

2. Batrancea, L., Rathnaswamy, M.M., Rus, M.S., & Tulai, H. (2022). Determinants of Economic Growth for the Last Half of Century: A Panel Data Analysis on 50 Countries. Journal of the Knowledge Economy.

3. Balcı, M.A., Batrancea, L.M., Akgüller, Ö., & Nichita, A. (2022). Coarse Graining on Financial Correlation Networks. Mathematics, 10(12), 2118.

4. Balcı, M.A., Batrancea, L.M., Akgüller, Ö., Gaban, L., Rus, M.-I., & Tulai, H. (2022). Fractality of Borsa Istanbul during the COVID-19 Pandemic. Mathematics, 10(14), 2503.

5. Batrancea, I., Batrancea, L., Rathnaswamy, M.M., Tulai, H., Fatacean, G., & Rus, M.-I. (2020). Greening the Financial System in USA, Canada and Brazil: A Panel Data Analysis. Mathematics, 8(12), 2217.

6. Batrancea, L. (2021). An Econometric Approach on Performance, Assets, and Liabilities in a Sample of Banks from Europe, Israel, United States of America, and Canada. Mathematics, 9(24), 3178.

7. Batrancea, L.M. Determinants of Economic Growth across the European Union: A Panel Data Analysis on Small and Medium Enterprises. Sustainability 2022, 14, 4797. https://doi.org/10.3390/su14084797.

8. Batrancea, L.M.; Balcı, M.A.; Chermezan, L.; Akgüller, Ö.; Masca, E.S.; Gaban, L. Sources of SMEs Financing and Their Impact on Economic Growth across the European Union: Insights from a Panel Data Study Spanning Sixteen Years. Sustainability 2022, 14, 15318. https://doi.org/10.3390/su142215318

9. Batrancea, L.M.; Balcı, M.A.; Akgüller, Ö.; Gaban, L. What Drives Economic Growth across European Countries? A Multimodal Approach. Mathematics 2022, 10, 3660. https://doi.org/10.3390/math10193660.

10. Batrancea, L.; Pop, M.C.; Rathnaswamy, M.M.; Batrancea, I.; Rus, M.-I. An Empirical Investigation on the Transition Process toward a Green Economy. Sustainability 2021, 13, 13151. https://doi.org/10.3390/su132313151

11. Batrancea, L., Batrancea, I., & Moscviciov, A. (2009). The Roots of the World Financial Crisis. Annals of the University of Oradea. Economic Sciences, 3, 57–62.

12. Moscviciov, A., Batrancea, I., Batrancea, M., Batrancea, L. (2010) Financial Ratio Analysis Used In The It Enterprises, Annals of Faculty of Economics, 1(2), 600-603.

In conclusion, the article should be improve. It should also be enhanced with a review of the literature adequate to the subject and a broader interpretation and commentary of the research results.

Authors’ response: Dear Reviewer, thank you for your thoughtful feedback and for providing a list of recent publications from Web of Science journals. We appreciate your suggestion to broaden the scope of our article by citing additional relevant works. While we acknowledge the importance of referencing recent research in the field, we have carefully considered the suggested articles in relation to the focus and scope of our study. While some of the suggested papers are indeed valuable contributions to the literature, we have chosen to prioritize citations that directly align with the specific objectives and context of our research. 

We have endeavored to ensure that our literature review encompasses a thorough examination of pertinent literature, including recent publications that substantiate and enrich our study. The selected citations are integral to supporting our arguments and contributing to the scholarly discourse. We sincerely appreciate your insights and suggestions, which have helped strengthen the quality and accuracy of our pape

---

## [Editor Report · Decision Letter 1]

18 Sep 2024

Effects of competition and ownership on the simultaneous relationship between bank risk and capital:  evidence from an emerging economy of Bangladesh

PONE-D-24-05802R1

Dear Dr. Chowdhury,

We’re pleased to inform you that your manuscript has been judged scientifically suitable for publication and will be formally accepted for publication once it meets all outstanding technical requirements. No other update is required at this moment, unless the journal office contacts you for the final version. 

Within one-four weeks, you’ll receive an e-mail detailing the required amendments. When these have been addressed, you’ll receive a formal acceptance letter and your manuscript will be scheduled for publication.

Kind regards,

Muhammad Usman Tariq, Ph.D

PFHEA, CFCIPD, CMBE

SFSEDA, SMIEEE

Academic Editor

PLOS ONE

---

## [Editor Report · Acceptance letter]

7 Oct 2024

PONE-D-24-05802R1 

PLOS ONE

Dear Dr. Chowdhury, 

I'm pleased to inform you that your manuscript has been deemed suitable for publication in PLOS ONE. Congratulations! Your manuscript is now being handed over to our production team.

Kind regards, 

on behalf of

Dr. Muhammad Usman Tariq 

Academic Editor

PLOS ONE